# Vasoactive intestinal peptide-expressing interneurons are impaired in a mouse model of Dravet syndrome

Kevin M Goff[1,2], Ethan M Goldberg[1,3,4]*

[1]Department of Neuroscience, The University of Pennsylvania Perelman School of Medicine, Philadelphia, United States; [2]The Medical Scientist Training Program, The University of Pennsylvania Perelman School of Medicine, Philadelphia, United States; [3]Division of Neurology, Department of Pediatrics, The Children's Hospital of Philadelphia, Philadelphia, United States; [4]Department of Neurology, The University of Pennsylvania Perelman School of Medicine, Philadelphia, United States

**Abstract** Dravet Syndrome (DS) is a severe neurodevelopmental disorder caused by pathogenic loss of function variants in the gene *SCN1A* which encodes the voltage gated sodium (Na$^+$) channel subunit Nav1.1. GABAergic interneurons expressing parvalbumin (PV-INs) and somatostatin (SST-INs) exhibit impaired excitability in DS (*Scn1a*$^{+/-}$) mice. However, the function of a third major class of interneurons in DS – those expressing vasoactive intestinal peptide (VIP-IN) –is unknown. We recorded VIP-INs in brain slices from *Scn1a*$^{+/-}$ mice and wild-type littermate controls and found prominent impairment of irregular spiking (IS), but not continuous adapting (CA) VIP-INs, in *Scn1a*$^{+/-}$ mice. Application of the Nav1.1-specific toxin Hm1a rescued the observed deficits. The IS vs. CA firing pattern is determined by expression of KCNQ channels; IS VIP-INs switched to tonic firing with both pharmacologic blockade of M-current and muscarinic acetylcholine receptor activation. These results show that VIP-INs express Nav1.1 and are dysfunctional in DS, which may contribute to DS pathogenesis.

DOI: https://doi.org/10.7554/eLife.46846.001

*For correspondence:
goldberge@email.chop.edu

**Competing interests:** The authors declare that no competing interests exist.

## Introduction

Dravet syndrome (DS) is a severe neurodevelopmental disorder characterized by temperature-sensitive and spontaneous seizures in the first year of life followed by progression to intractable epilepsy, intellectual disability with autistic features, and a markedly increased rate of sudden unexpected death (SUDEP) (*Genton et al., 2011*; *Kalume et al., 2013*; *Oakley et al., 2009*; *Scheffer, 2012*). DS is caused by pathogenic loss of function variants in the gene *SCN1A*, resulting in haploinsufficiency of the voltage gated sodium (Na$^+$) channel α subunit Nav1.1 (*Claes et al., 2001*). Experimental models of DS, including mice lacking a functional copy of Nav1.1 (*Scn1a*$^{+/-}$ mice) (*Mistry et al., 2014*; *Ogiwara et al., 2007*; *Yu et al., 2006*) and neurons generated from induced pluripotent stem (iPS) cells derived from DS patients (*Liu et al., 2013*; *Sun et al., 2016*), have provided important insights into the cellular and circuit basis of this disorder. Parvalbumin-immunopositive fast-spiking GABAergic interneurons (PV-INs) and somatostatin (SST)-expressing dendrite-targeting INs (SST-INs) exhibit impaired excitability in *Scn1a*$^{+/-}$ mice (*Favero et al., 2018*; *Ogiwara et al., 2007*; *De Stasi et al., 2016*; *Tai et al., 2014*; *Yu et al., 2006*). Despite advances in the understanding of underlying disease pathogenesis, DS remains incurable with a poor prognosis. How loss of Nav1.1 leads to the features that define this clinical syndrome is incompletely understood, and identifying new, targetable loci of dysfunction is critically important to developing new treatments.

Nav1.1 is expressed at the axon initial segment (AIS) of PV-INs and, to a lesser extent, SST-INs, and is a major determinant of action potential (AP) generation in these cells (*Li et al., 2014*; *Ogiwara et al., 2007*; *De Stasi et al., 2016*; *Tai et al., 2014*). Such data have led to the 'interneuron hypothesis' of DS, which posits that PV and SST-IN dysfunction is the predominant driver of both epilepsy and autism (*Catterall, 2018*; *Catterall et al., 2010*; *Han et al., 2012*; *Ogiwara et al., 2007*; *Rubinstein et al., 2015*; *Yu et al., 2006*). However, we recently showed that PV-IN hypoexcitability in $Scn1a^{+/-}$ mice is restricted to an early and transient developmental window (P11-21), with subsequent normalization of high-frequency action potential discharge by P35 (*Favero et al., 2018*). This finding encourages a reconsideration of the cellular basis of circuit pathology that underlies chronic epilepsy and epilepsy-associated cognitive abnormalities in DS. The functional status of the rich diversity of interneurons in the cerebral cortex (*DeFelipe et al., 2013*; *Tremblay et al., 2016*) has not been fully explored in models of DS. It could be the case that other interneuron classes are also affected and exhibit chronic dysfunction. The impact of constitutive loss of Nav1.1 on the third major class of cortical INs – those expressing vasoactive intestinal peptide (VIP-INs) – remains unknown.

VIP-INs are functionally distinct from SST and PV-INs as they preferentially target other interneurons (particularly SST-INs) rather than pyramidal cells, and thus are considered to exert a disinhibitory influence on cerebral cortical circuits (*Acsády et al., 1998*; *Fu et al., 2014*; *Krabbe et al., 2018*; *Lee et al., 2013*; *Muñoz et al., 2017*; *Pi et al., 2013*; *Turi et al., 2019*; *Zhang et al., 2014*). VIP-INs also have a role in regulating the response of the cerebral cortex to ascending cholinergic neuromodulatory input from the basal forebrain (*Alitto and Dan, 2012*; *Fu et al., 2015*; *Kawaguchi, 1997*; *Porter et al., 1999*), and hence are involved in higher-order functions such as attention, memory, and cognitive processing, all of which are impaired in DS (*Dravet and Oguni, 2013*). Prior work has suggested that VIP-INs do not express Nav1.1 (*Yamagata et al., 2017*), while recent single cell RNA sequencing studies do suggest expression of Nav1.1 in VIP-INs (*Paul et al., 2017*; *Tasic et al., 2016*). Here, we assessed the functional status of VIP-INs in $Scn1a^{+/-}$ mice and directly investigated whether VIP-INs express Nav1.1.

To address this question, we performed targeted whole cell recordings from layer 2/3 VIP-INs in acute brain slices of primary somatosensory and visual cortex in male and female $Scn1a^{+/-}$ mice and age-matched wild-type (WT) littermate controls at two different developmental time points. We found that VIP-INs exhibit impairment of AP generation and repetitive firing and confirmed the presence of Nav1.1 on putative VIP-IN axons using immunohistochemistry and confocal microscopy. However, unlike the transient abnormalities seen in PV-INs (*Favero et al., 2018*), this VIP-IN deficit persisted at later developmental time points. We further found that VIP-IN dysfunction in $Scn1a^{+/-}$- mice was restricted to a subset of irregular spiking (IS) VIP-INs, while continuous adapting (CA) VIP-INs were essentially normal. Deficits seen in IS VIP-INs could be reversed by application of the peptide toxin Hm1a, a recently-identified Nav1.1-specific modulator (*Osteen et al., 2016*). The basis of the IS firing pattern involved M-current mediated by KCNQ channels likely containing Kcnq5. Blockade of M-current with the KCNQ channel inhibitor linopirdine converted IS VIP-INs to a tonic firing mode but had no effect on CA VIP-INs. Also, muscarinic, but not nicotinic cholinergic receptor activation reversibly converted IS VIP-INs to a continuous firing mode. Overall, these results identify a novel cellular locus of dysfunction in DS that may relate to the profound and durable cognitive impairments observed in human patients (*Genton et al., 2011*; *Scheffer, 2012*; *Villas et al., 2017*) and additionally refines our understanding of the diversity of VIP-INs and the molecular determinants of VIP-IN function.

## Results

### Neocortical vasoactive intestinal peptide expressing interneurons exhibit impaired excitability in $Scn1a^{+/-}$ mice

To assess the function of neocortical VIP-INs in $Scn1a^{+/-}$ mice, we performed targeted whole cell patch clamp recordings from fluorescently labeled neurons in layer 2/3 of primary somatosensory ('barrel') cortex in acute brain slices prepared from juvenile and young adult triple transgenic *Scn1a. VIP-Cre.tdTomato* (tdT) mice and age-matched WT.VIP-Cre.tdT littermate controls (see Materials and methods). VIP-INs from $Scn1a^{+/-}$ mice demonstrated multiple abnormalities that were consistent

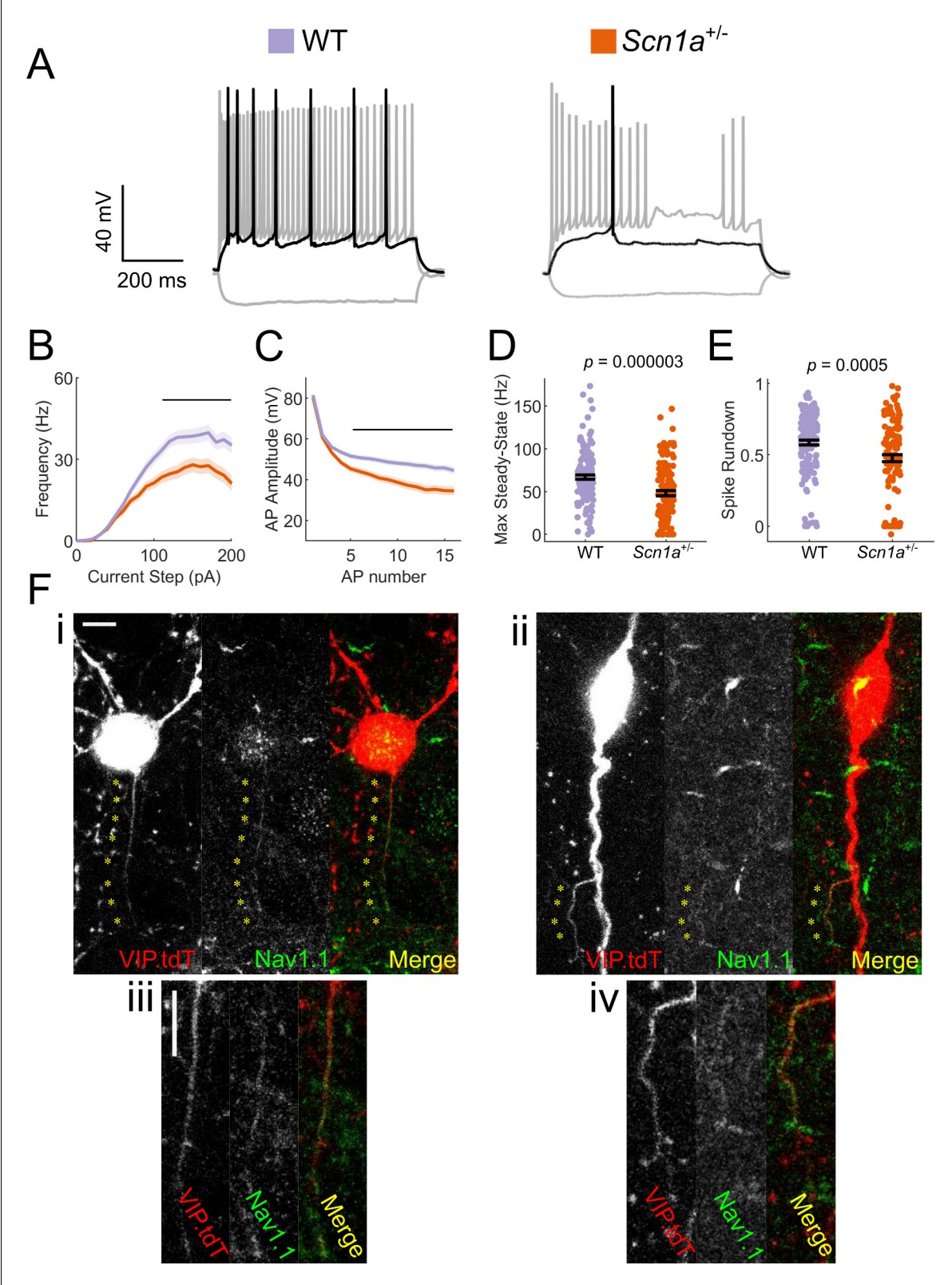

**Figure 1.** VIP-INs express Nav1.1 and are hypoexcitable in *Scn1a*[+/-] mice. (**A**) Representative traces of layer 2/3 VIP-INs from a *Scn1a*[+/-] mouse and age matched WT littermate. Location of scale bar indicates −70 mV. *Black* indicates rheobase current injection, while *gray* indicates response to a −50 pA hyperpolarizing and a 2X rheobase suprathreshold current injection. (**B**) Current/frequency (*I*-f) output curve of *n* = 123 VIP-INs from 24 *Scn1a*[+/-] mice and *n* = 150 VIP-INs from 38 age matched WT littermates (See ***Figure 3—figure supplement 1*** for age specific data). Line and shaded area represent

*Figure 1 continued*

mean ± SEM, and the black bar indicates significance at p<0.01 by a multivariate ANOVA and post-hoc Bonferroni correction. (C) Population average of spike amplitude (AP peak – AP threshold) of repeated action potentials measured at 3X rheobase for each cell. Line and shaded area represent mean ± SEM. The black bar indicates significance at p<0.01 by a multivariate ANOVA and post-hoc Bonferroni correction. (D) Bar graphs showing individual cell values (dots) and the population mean ± SEM for the maximal steady-state firing frequency of each cell (see Materials and methods). *p* values determined by Mann–Whitney U test. (E) Bar graphs showing individual cell values (dots) and the population mean ± SEM for the reduction in spike amplitude of the tenth AP measured in B normalized to the first AP. *p* values determined by Mann–Whitney U test. All comparisons consider each cell as n = 1 (see *Table 1* for per-animal comparisons). (F) Example images of WT VIP-INs showing immunohistochemical labeling of Nav1.1 on putative axons originating either from the soma (i, inset iii) or a proximal dendrite (ii, inset iv). Asterisks indicate regions of colocalization between Nav1.1 and tdT expression in VIP-INs. Scale is 5 μm.

DOI: https://doi.org/10.7554/eLife.46846.002

The following figure supplements are available for figure 1:

**Figure supplement 1.** *Scn1a* expression in VIP-INs.
DOI: https://doi.org/10.7554/eLife.46846.003
**Figure supplement 2.** Localization of Nav1.1 on VIP-IN axons.
DOI: https://doi.org/10.7554/eLife.46846.004
**Figure supplement 3.** Validation of Nav1.1 staining.
DOI: https://doi.org/10.7554/eLife.46846.005

with loss of $Na^+$ current (*Figure 1*), including a reduced maximal steady-state firing frequency (67 ± 2 vs. 48 ± 3 Hz; p=3 X E-6; *n* = 150/123 WT/*Scn1a$^{+/-}$*), as well as a marked shift in the current/frequency (I/f; input-output) curve. There was notable spike height accommodation (rundown) during repetitive action potential discharge with a progressive depolarization of AP threshold and a decrease in AP peak value during suprathreshold current injections, suggesting enhanced accumulation of $Na^+$ channel inactivation in VIP-INs from *Scn1a$^{+/-}$* mice. These data support the conclusion that neocortical VIP-INs are hypoexcitable in *Scn1a$^{+/-}$* mice relative to age-matched WT littermate controls, likely due to absence of one copy of *Scn1a* and a resulting decrease in $Na^+$ current.

## VIP interneurons express Nav1.1

Nav1.1 is known to be expressed in PV-INs and SST-INs, while a recent study suggested that VIP-INs express Nav1.2 at higher levels (*Yamagata et al., 2017*). However, several recent large-scale transcriptomics datasets suggest that *Scn1a* mRNA is in fact expressed in VIP-INs, and at levels similar to that seen in SST-INs (*Figure 1—figure supplement 1*) (*Paul et al., 2017*; *Tasic et al., 2016*). We performed immunohistochemical analysis to confirm the presence of Nav1.1 protein in VIP-INs. We identified fine (0.5 μm) tdT positive, Nav1.1-immunoreactive processes emanating from the soma (*Figure 1Ei–ii*) or proximal dendrite (*Figure 1Eiii-iv*) of labeled neurons in WT.VIP-Cre.tdT mice; these were considered to correspond to the VIP-IN axon based on small caliber and larger branching angle (*Prönneke et al., 2015*). We identified the axon in 30 VIP-INs and found that 23 of 30 (77%) identified axons expressed Nav1.1 (*Figure 1—figure supplement 2*). In order to validate the specificity of this staining, we performed Nav1.1 immunohistochemistry of tissue from *Scn1a$^{-/-}$* null mice and found no immunoreactivity (*Figure 1—figure supplement 3A–B*). As a positive control, we found high expression of Nav1.1 on PV-IN axons (*Figure 1—figure supplement 3C*). These data confirm that Nav1.1 protein is present on the axon of many VIP-INs.

## Two electrophysiological subgroups of VIP-INs in neocortical layer 2/3

Results presented thus far indicate that VIP-INs express Nav1.1 and exhibit impaired excitability in *Scn1a$^{+/-}$* mice. In the course of these initial experiments, we observed a subset (~50%) of VIP-INs in *Scn1a$^{+/-}$* mice that only fired a brief train of APs before complete cessation of firing (*Figure 1A*), which was atypical for VIP-INs from WT mice. This prompted us to explore how our data from *Scn1a$^{+/-}$* mice mapped onto previously described VIP-IN firing patterns. VIP-INs in superficial layers of mouse and rat barrel cortex exhibit spike frequency adaptation with or without irregular spiking and/or initial bursting (variably referred to in the literature as continuous adapting, irregular spiking, bursting, or fast adapting) (*He et al., 2016*; *Prönneke et al., 2015*). However, there is no existing standardized or widely agreed upon nomenclature to describe VIP-IN firing patterns.

We identified a diversity of VIP-IN firing patterns in response to step depolarizations consistent with previously described continuous adapting (CA), irregular spiking (IS), and bursting cells (*von Engelhardt et al., 2007*; *He et al., 2016*; *Lee et al., 2010*; *Porter et al., 1998*; *Porter et al., 1999*; *Prönneke et al., 2015*). However, when we delivered longer (8–10 s) suprathreshold depolarizing pulses, we found that all cells could be separated into two clear groups (*Figure 2*). Approximately half of all VIP-INs fired continuously, albeit with spike frequency adaptation (here referred to as CA VIP-INs), while the other half instead fired a burst of action potentials prior to switching to an irregular spiking pattern (IS VIP-INs), often with a variable silent period in between the two phases (*Figure 2A*). We could reliably identify these two groups using k-means clustering of the length of the initial burst of action potentials and the coefficient of variation of the inter-spike interval (ISI CoV), but these groups were not distinguishable when using 600 ms pulses alone (*Figure 2C,D*). Many cells that fired continuously during 600 ms sweeps were in fact clearly IS based on response to these longer depolarizations; likewise, cells that might be classified as bursting and which appear to cease firing during 600 ms sweeps eventually switch to an IS mode with longer depolarizations.

A recent study implicated T-type calcium channels as the mechanism of bursting at rheobase observed in a small subset (~20%) of VIP-INs (*Prönneke et al., 2018*). We considered whether this group corresponded to the IS VIP-INs described here. A prepulse step depolarization to −55 mV to inactivate T-type channels had no effect on the suprathreshold response of IS-VIPs here (*Figure 2— figure supplement 1A–B*). We were able to clearly identify CA and IS VIP-INs using a slow 8 s ramp current injection to induce inactivation of T-type calcium channels (*Figure 2—figure supplement 1C–E*). Finally, we found that a subset of both CA and IS VIP-INs burst at rheobase, indicating that our IS VIP-IN category does not correspond directly to these previously described bursting VIP-INs (*Figure 2—figure supplement 2*). Hence, we concluded that irregular spiking vs. bursting at rheobase are mechanistically distinct features of in VIP-INs.

## IS VIP-INs are specifically impaired in *Scn1a*$^{+/-}$ mice

While VIP-INs exhibit decreased excitability in *Scn1a*$^{+/-}$ mice, dividing the dataset into CA and IS VIP-INs revealed that IS VIP-INs are particularly impaired while CA VIP-INs are relatively spared. We determined that CA and IS VIP-INs are present in *Scn1a*$^{+/-}$ and WT mice in similar proportions as determined by our unbiased classification (*Figure 2B,E*). IS VIP-INs displayed more severe deficits compared to the pooled dataset (*Figure 1*), with a depolarized AP threshold (WT vs. *Scn1a*$^{+/-}$ mice: −41.3 ± 0.5 vs. −39.3 ± 0.4; p=0.01), decreased AP amplitude (79 ± 1.0 vs. 72 ± 1.3 mV; p=0.003) and steady-state firing (59 ± 4 vs. 25 ± 3 Hz; p=8 X E-9), a clear difference in the *I*-f curve, and notable spike rundown with repetitive discharge (*Figure 3*). These differences remained significant when we treated each animal (rather than a cell) as an *n* (*Table 1*). CA VIP-INs from *Scn1a*$^{+/-}$ mice showed no statistical differences compared to CA VIP-INs from age-matched WT littermate controls across a range of additional measures of intrinsic excitability, properties of individual action potentials, and features of repetitive firing (*Table 1*). Additionally, subtype differences between CA and IS VIP-INs were consistent between WT and *Scn1a*$^{+/-}$ mice, including presence of increased voltage sag in response to hyperpolarizing current injection and shorter (faster) AP half-width and rise-time, with IS VIP-INs more likely to burst at rheobase (*Figure 2—figure supplement 2*, *Table 1*).

As prior work showed that the electrophysiological abnormalities identified in PV-INs in *Scn1a*$^{+/-}$-mice are limited to a narrow developmental window between P11-21 (*Favero et al., 2018*), we subdivided our analysis based on age and found nearly identical results at P18-21 and P30-55 for both IS and CA VIP-INs (*Figure 3—figure supplement 1*). To assess whether impairment of IS VIP-INs was a general phenomenon or restricted to primary somatosensory neocortex, we additionally performed a set of recordings in layer 2/3 primary visual cortex, again finding a similar proportion of IS vs. CA VIP-INs and replicating our central finding that IS VIP-INs were selectively hypoexcitable in *Scn1a*$^{+/-}$ mice (*Figure 3—figure supplement 2*).

## Modulation of Nav1.1 rescues IS VIP-IN function in *Scn1a*$^{+/-}$ mice

To further support our hypothesis that VIP-INs are impaired in *Scn1a*$^{+/-}$ mice due to decreased expression of Nav1.1, we attempted to rescue VIP-IN function by directly targeting Nav1.1 pharmacologically. Hm1a is a recently described spider toxin that enhances Nav1.1 current via inhibiting fast and slow inactivation (*Osteen et al., 2016*; *Osteen et al., 2017*). In acute brain slices prepared from

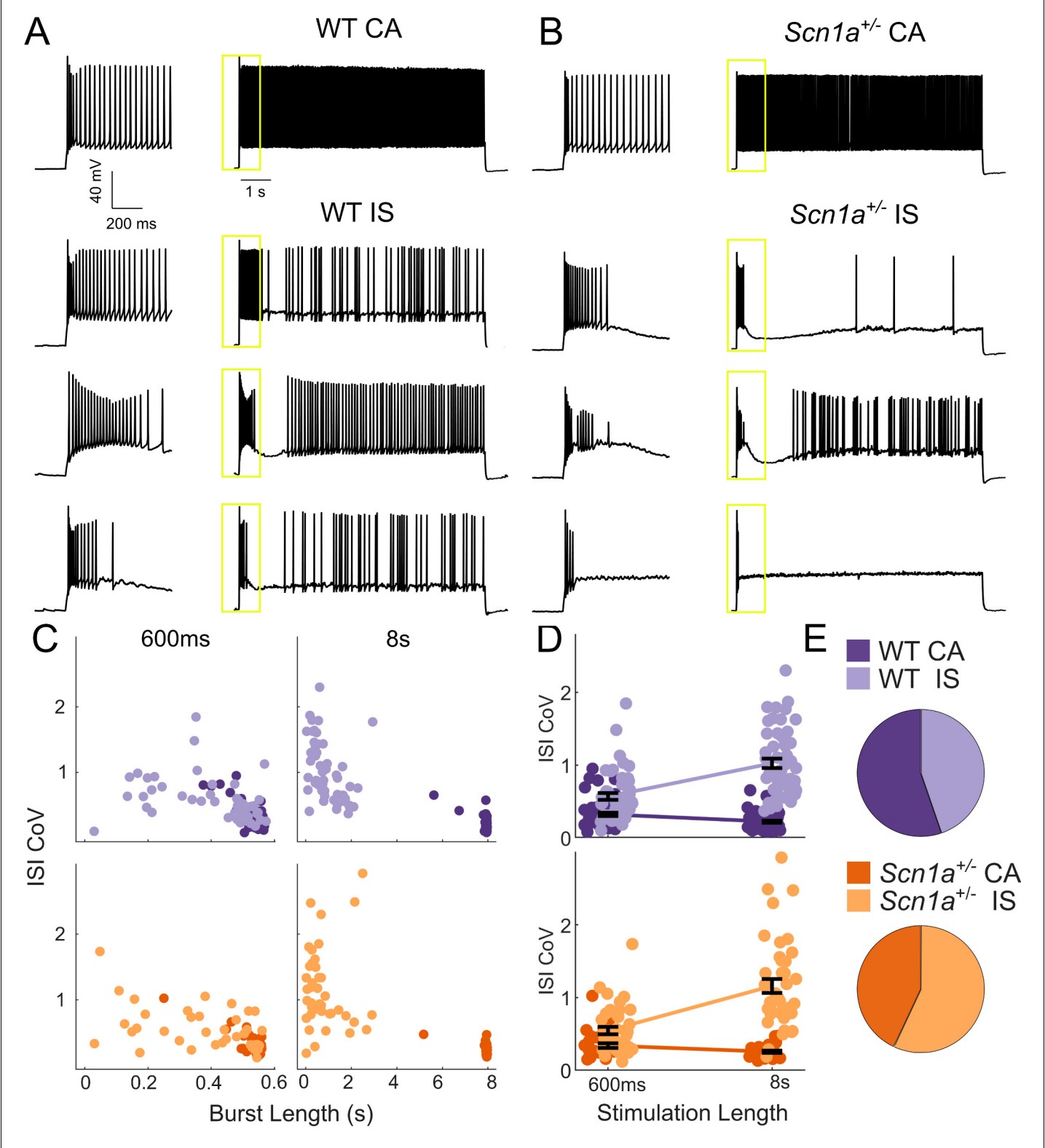

**Figure 2.** Two VIP-IN firing patterns revealed with long depolarization. (**A**) Four representative cells displaying the range of VIP-IN firing patterns. On the left are APs elicited with a standard (600 ms) current injection. On the right are firing patterns recorded with 8 s long depolarizations using the same current amplitude as on the left (2X rheobase for each cell). CA and IS firing patterns are easily separated with 8 s depolarization, but not distinguishable using only the first 600 ms. These firing patterns were consistent even when we stimulated with higher current injections (see *Figure 3Aii–iii*). (**B**) Similar CA and IS patterns were observed in VIP-INs from *Scn1a*[+/-] mice, with a deficit in IS VIP-IN firing. (**C**) Characterization of VIP-

*Figure 2 continued on next page*

*Figure 2 continued*

IN firing patterns using two key measures: the length of the initial burst of action potentials, and the coefficient of variation of the inter-spike interval (ISI CoV). These parameters were measured using either 600 ms or 8 s long rectangular depolarization in cells from both WT and *Scn1a*$^{+/-}$ mice; then, k-means clustering of the 8 s data was performed (light/dark colors indicating the results of clustering into two groups). Note the clear separation between the two groups in the 8 s data, but large overlap when looking only at data generated from 600 ms current injections. (D) The same data as in C, but highlighting the dependence of the measured ISI CoV on the length of the depolarization. With 600 ms sweeps, there is considerable overlap between CA and IS cells in both genotypes; however, with 8 s depolarizations, the two groups separate with minimal overlap. (E) Similar proportions of CA and IS cells in WT and *Scn1a*$^{+/-}$ mice. For C-E, *n* = 51 IS and 63 CA from 24 WT mice, and *n* = 41 IS and 31 CA from 17 *Scn1a*$^{+/-}$ mice; p=0.10 via chi-square test; chi-square statistic = 2.63.

DOI: https://doi.org/10.7554/eLife.46846.006

The following figure supplements are available for figure 2:

**Figure supplement 1.** IS VIP-IN firing patterns are robust to a variety of stimulation methods.

DOI: https://doi.org/10.7554/eLife.46846.007

**Figure supplement 2.** A subset of both CA and IS VIP-INs show bursting at rheobase.

DOI: https://doi.org/10.7554/eLife.46846.008

*Scn1a*$^{+/-}$ mice, bath application of 1 µM Hm1a led to markedly increased steady-state firing in IS VIP-INs (control, 33 ± 2 Hz; Hm1a, 97 ± 5 Hz; p=0.001) and attenuation of spike rundown (*Figure 4A–C*). We did observe an effect in CA VIP-INs at this concentration, with a small increase in steady-state firing frequency (control, 58 ± 4 Hz; Hm1a, 68 ± 5 Hz; p=0.05) and attenuated spike height accommodation. As a positive control, Hm1a increased the maximum steady-state firing rate of PV-INs in *Scn1a*$^{+/-}$ mice at P18-21 (control, 229 ± 2 Hz; Hm1a, 292 ± 6 Hz; p=0.004), although this effect was proportionally smaller than what we observed in IS VIP INs (mean difference of 295% in IS VIP-INs compared to 28% in PV-INs). In contrast, Hm1a had no measurable effect on the firing of layer 2/3 neocortical pyramidal neurons in WT mice, supporting the specificity of this toxin to Nav1.1 and the smaller role of Nav1.1 in the regulation of pyramidal cell excitability relative to interneurons. We repeated this rescue experiment of IS VIP-IN excitability using 50 nM Hm1a, which is highly selective for Nav1.1 over other non-Nav1.1 Nav1.X subunits (*Osteen et al., 2016*; *Richards et al., 2018*), and obtained identical results (*Figure 4—figure supplement 1*). Importantly, although Hm1a causes a dramatic increase in steady-state firing of IS VIP-INs, these cells still retain a distinct irregular firing pattern in the presence of Hm1a in response to longer depolarizations (*Figure 4—figure supplement 1*).

## Hm1a modulates Na$^+$ current inactivation in VIP-INs

The mechanism of action of Hm1a involves destabilizing the inactive state of Nav1.1, which increases the availability of Nav1.1 channels at depolarized potentials by shifting more channels to the closed vs. inactive state. If VIP-INs indeed express Nav.1, then Hm1a could lead to a rescue of IS VIP-IN function by preventing an accumulation of Na$^+$ channel inactivation during repetitive firing. To test this possibility, we directly measured the effect of Hm1a on Na$^+$ current in VIP-INs using whole cell voltage clamp recordings of genetically labeled VIP-INs isolated from acutely dissociated cortex of P18 WT.VIP-Cre.tdT mice (*Figure 4D–F*). Hm1a slowed the time constant of inactivation and increased slowly-inactivating/persistent current in VIP-INs, with no effect on the peak current density. This further supports the presence of Nav1.1 in VIP-INs and indicates that Hm1a has a similar effect on inactivation of endogenous Na$^+$ current in VIP-INs as compared to previously published data on Nav1.1 expressed in heterologous systems.

## Irregular spiking is not correlated with VIP-IN molecular markers

The rescue of IS VIP-IN excitability with Hm1a supports our conclusion that IS VIP-INs are particularly impaired in *Scn1a*$^{+/-}$ mice. To attempt to understand how the specific dysfunction of IS VIP-INs is involved in DS pathogenesis, we investigated whether the IS and CA firing patterns identified here map onto previously described morphological and molecularly defined VIP-IN subgroups. Intersectional Cre/Flp labeling of VIP/cholecystokinin (VIP/CCK) or VIP/calretinin (VIP/CR) double positive INs defines two minimally-overlapping subsets of VIP-INs (*He et al., 2016*; *Porter et al., 1998*). In these mice, VIP/CR-INs have a bipolar morphology, while VIP/CCK INs tend to be multipolar. We used whole cell recording with morphological analysis to compare our electrophysiological

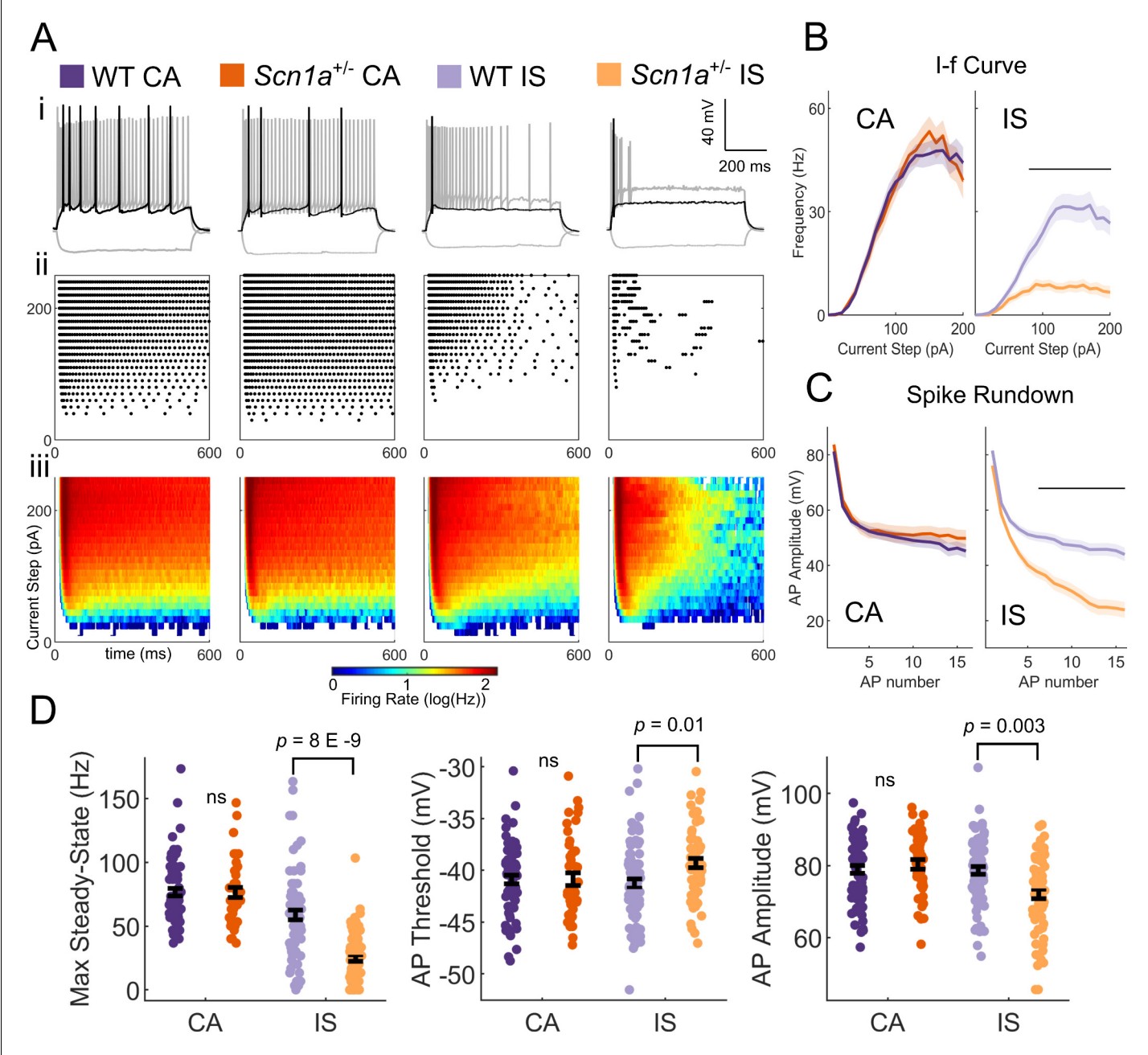

**Figure 3.** IS VIP-INs are preferentially impaired in *Scn1a*[+/-] mice. (**A**) (**i**) Example traces of CA and IS VIP-INs from WT and *Scn1a*[+/-] mice, showing responses at −50 pA hyperpolarizing (*gray*), rheobase (*black*), and 2X rheobase current injections (*gray*; typically ~ 100–140 pA). (**ii**) Raster plots of all APs elicited with standard 600 ms current steps in 10 pA increments for the cells in i. (**iii**) Population averages of the raster plots in ii, represented as heat maps where the color corresponds to the instantaneous firing rate for each give current step using a 20 ms sliding average. (**B**) Current/frequency (IF) plots for groups shown in A. Line and shaded area represent mean ± SEM. Bar indicates significance at p<0.001 by a multivariate ANOVA and post-hoc Bonferroni correction. (**C**) Population average of AP amplitude for repeated action potentials elicited at 3X rheobase for each cell. Line and shaded area represent mean ± SEM. Bar indicates significance at p<0.001 by a multivariate ANOVA and post-hoc Bonferroni correction. (**D**) Bar graphs showing individual cell values (dots) and the population mean ± SEM for several markers of VIP-IN excitability. *p* values determined by Kruskal-Wallis with post-hoc Dunn's test. For simplicity, differences between CA an IS cells are not shown (see *Table 1*). For B-D, n = 71 IS and 78 CA cells from 32 WT mice, and n = 66 IS and 43 CA cells from 22 *Scn1a*[+/-] mice were included. All groups were determined using 8 s square (*Figure 2*) or ramp depolarizations (*Figure 2—figure supplement 1*) or, for a subset of IS VIP-INs for which we did not record 8 s depolarizations, if their definitive firing pattern was clear with a 600 ms depolarization (as in Ai). Data for each cell are available in *Figure 3—source data 1*.

DOI: https://doi.org/10.7554/eLife.46846.009

*Figure 3 continued on next page*

*Figure 3 continued*

The following source data and figure supplements are available for figure 3:

**Source data 1.** VIP-IN neurophysiological data.
DOI: https://doi.org/10.7554/eLife.46846.012
**Figure supplement 1.** IS VIP-IN deficits in *Scn1a*$^{+/-}$ mice are consistent across development.
DOI: https://doi.org/10.7554/eLife.46846.010
**Figure supplement 2.** IS VIP-IN deficits in *Scn1a*$^{+/-}$ mice are consistent across cortical areas.
DOI: https://doi.org/10.7554/eLife.46846.011

classification scheme with the subtypes labeled via this intersectional strategy (*Figure 5*). We filled cells with Alexa-488 during whole-cell recording, generated 2P image stacks, and quantified the dendritic arborization pattern of each cell. VIP/CR-INs were more often bipolar, or 'vertically biased,' compared to VIP/CCK-INs (*Figure 5A*), as shown previously (*He et al., 2016*), but CR and CCK did not reliably define electrophysiological subtypes of VIP-INs (*Figure 5A–B*), nor were there any differences in morphology between the CA and IS VIP-INs within a given genotype (*Figure 5C*). These data argue that IS and CA firing patterns do not coincide with previously described VIP-IN subgroups.

## Irregular spiking of VIP-INs is determined by M-current

Despite the negative finding above, we were interested in the molecular determinants of the IS discharge pattern to better understand why IS VIP-INs are selectively impaired in *Scn1a*$^{+/-}$ mice. The firing pattern of IS VIP-INs could suggest the presence of a slowly activating potassium conductance in

**Table 1.** Properties of VIP-INs from *Scn1a*$^{+/-}$ and WT littermates Listed p-values indicate the result of a Kruskal Wallis test for an effect by group.

Pairwise comparisons are with Dunn's test. All comparisons are made considering the average of each mouse as n = 1.* p<0.05 vs. age and subtype matched wild-type; \*\*p<0.01; \*\*\*p<0.001. † p<0.05 between CA and IS subtypes (consistent between WT and Scn1a$^{+/-}$). Data for each cell are reported in *Figure 3—source data 1*.

| Subtype | CA | | | IS | | | p-value |
|---|---|---|---|---|---|---|---|
| Genotype | WT | Scn1a$^{+/-}$ | | WT | | Scn1a$^{+/-}$ | (group) |
| *n* mice (cells) | 30(71) | 20(43) | | 32(78) | | 22(66) | |
| Age (days) | 33.7 ± 1.9 | 33.5 ± 2.6 | | 33.4 ± 2.0 | | 34.0 ± 2.4 | 0.8 |
| *Vm* (mV) | −66.3 ± 1.0 | −65.2 ± 1.0 | | −65.6 ± 0.8 | | −64.0 ± 0.6 | 0.3 |
| *Rm* (MΩ) | 355 ± 16 | 391 ± 21 | | 344 ± 17 | | 314 ± 11 | 0.2 |
| Time Constant | 9.17 ± 0.9 | 9.98 ± 1.0 | | 10.7 ± 1.5 | | 10.1 ± 0.9 | 0.2 |
| Rheobase (pA) | 48.3 ± 3.2 | 45.4 ± 4.1 | | 53.5 ± 3.6 | | 57.7 ± 4.2 | 0.2 |
| AP Threshold (mV) | −41.8 ± 0.5 | −41.5 ± 0.5 | | −41.2 ± 0.5 | * | −39.1 ± 0.4 | 0.01 |
| AP Rise Time (ms) | 0.45 ± 0.01 | 0.46 ± 0.01 | | 0.44 ± 0.01 | | 0.44 ± 0.01 | 0.1 |
| Max Rise Slope (mV/ms) | 418 ± 14 | 417 ± 13 | | 427 ± 17 | | 371 ± 15 | 0.1 |
| AP Halfwidth (ms) | 0.55 ± 0.02 | 0.54 ± 0.02 | † | 0.50 ± 0.02 | | 0.49 ± 0.02 | 0.04 |
| AP Amplitude (mV) | 79 ± 1.3 | 80 ± 1.2 | | 78 ± 1.6 | * | 71 ± 1.8 | 0.004 |
| AHP Amplitude (mV) | 10.0 ± 0.63 | 10.9 ± 0.55 | | 10.6 ± 0.55 | | 11.0 ± 0.43 | 0.6 |
| AHP time (ms) | 1.75 ± 0.16 | 1.76 ± 0.21 | | 1.51 ± 0.14 | | 1.41 ± 0.11 | 0.09 |
| Sag (percent) | 14.4 ± 1.3 | 16.3 ± 1.5 | † | 23.5 ± 2.6 | | 28.0 ± 2.0 | 8 E −5 |
| APs at Rheobase | 1.38 ± 0.1 | 1.32 ± 0.1 | † | 2.41 ± 0.3 | | 2.01 ± 0.2 | 7 E −5 |
| Instantaneous (Hz) | 237 ± 11 | 235 ± 13 | | 245 ± 10 | | 209 ± 11 | 0.1 |
| Steady-State (Hz) | 81 ± 4 | 81 ± 4 | † | 55 ± 4 | ** | 25 ± 3 | 2 E −12 |
| ISI CoV | 0.30 ± 0.02 | 0.31 ± 0.02 | † | 0.61 ± 0.03 | | 0.69 ± 0.04 | 4 E −13 |

DOI: https://doi.org/10.7554/eLife.46846.013

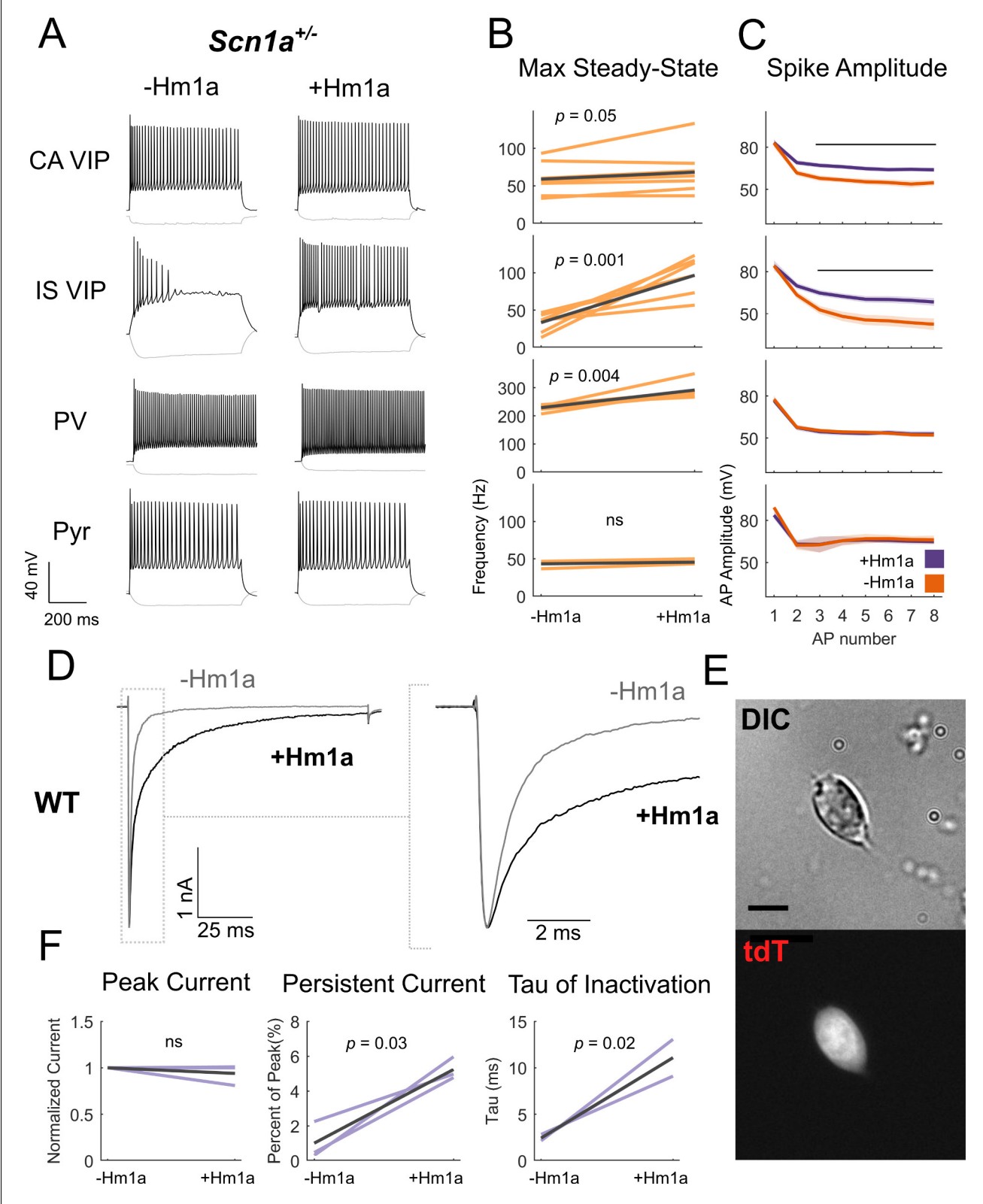

**Figure 4.** Hm1a application modulates Na$^+$ channels containing Nav1.1 subunits in VIP-INs and recovers IS VIP-IN hypofunction in *Scn1a*$^{+/-}$ mice.
(**A**) Example traces from recordings of representative CA and IS VIP-INs from *Scn1a*. VIP-Cre.tdT mice, as well as a PV-IN and pyramidal neuron from a *Scn1a*.PV-Cre.tdT mouse, before and after bath application of 1 μM Hm1a at a 3X rheobase (near maximal) current injection for each cell. Note that the horizontal scale for the PV-IN is 100 ms instead of 200 ms, to facilitate visualization of individual APs. (**B**) Change in the max steady state firing frequency

*Figure 4 continued on next page*

*Figure 4 continued*

of *n* = 9 CA VIP-INs, 7 IS VIP-INs, 6 PV-INs, and three pyramidal neurons from a total of 5 *Scn1a*.VIP-Cre.tdT and 2 *Scn1a*.PV-Cre.tdT P18-21 mice, with *p* values and significance determined using a paired students' t-test. (C) Spike amplitude of successive APs elicited at 3X rheobase for each cell. Line and shaded area represent mean ± SEM, and bar indicates significance at p<0.01 via a multivariate ANOVA and post-hoc Bonferroni correction. (D) Example traces from voltage clamp recordings of VIP-INs from acutely dissociated cortex of P18 WT.VIP-Cre.tdT mice. *Light gray* shows the initial transient sodium current recorded with a single voltage command step from −80 mV to 0 mV. *Black* shows the response following bath application of 500 nM Hm1a. The dashed line indicates the inset (shown on the *right*). There is no change in the peak amplitude, but clear slowing of inactivation. (E) Example differential interference contrast image of a dissociated VIP-IN, as well as the tdT signal imaged with epifluorescence. VIP-INs had small bipolar or rounded shapes. Scale = 5 μM. (F) Quantification of the effects of Hm1a on *n* = 3 VIP-INs from 2 P18 WT.VIP-Cre.tdT mice. Purple lines represent the change of each individual cell, with *p* values determined by a paired students' t-test.

DOI: https://doi.org/10.7554/eLife.46846.014

The following figure supplement is available for figure 4:

**Figure supplement 1.** Effects of low concentration Hm1a on IS VIP-IN firing.

DOI: https://doi.org/10.7554/eLife.46846.015

these cells. M-current, mediated by KCNQ channels, is known to possess such properties and has been hypothesized to regulate mode switching between tonic and bursting/irregular firing (*Drion et al., 2010*; *Stiefel et al., 2013*). Single-cell transcriptomics data suggest relatively high expression of the KCNQ subunit Kcnq5 in a subset of VIP-INs, including approximately half of all VIP/CR and VIP/CCK-INs, with minimal expression in either PV or SST-INs. KCNQ subfamily member Kcnq2 is more broadly expressed across all INs, while Kcnq1/3/4 show very limited cortical expression (*Figure 6—figure supplement 1*; *Paul et al., 2017*; *Tasic et al., 2016*). We confirmed these data using immunohistochemistry for Kcnq5 and Kcnq3. Kcnq5 was expressed in ~ 40% of layer 2/3 VIP-INs and essentially no SST-INs (but was expressed in many presumptive pyramidal cells), while Kcnq3 showed very limited immunoreactivity in the superficial cortex overall and was not found on VIP-INs (*Figure 6—figure supplement 2*).

Based on these data, we hypothesized that M-current mediated by Kcnq5-containing KCNQ channels underlies irregular spiking in IS VIP-INs, while CA VIP-INs do not rely on M-current for their firing properties. Using KCNQ channel-specific pharmacology, we were able to bidirectionally modulate the firing patterns of IS and CA VIP-INs, interconverting firing patterns by either blocking or activating M-current with the KCNQ channel inhibitor linopirdine or the KCNQ channel activator retigabine, respectively (*Figure 6*). IS VIP-INs showed a marked increase in sustained firing with the addition of linopirdine and resembled CA VIP-INs in terms of burst length and ISI CoV. Conversely, some CA VIP-INs became 'IS-like' upon application of retigabine. However, linopirdine had essentially no effect on CA VIP-IN excitability or intrinsic properties (*Figure 6—figure supplement 3*) indicating that whatever M-current is present in these cells is not a major regulator of cellular excitability. Such data indicate that M-current expression is a critical but selective determinant of the electrical excitability of IS VIP-INs. However, M-current does not seem to correlate with expression of VIP-IN markers such as CR and CCK, or with VIP-IN morphology.

## Cholinergic switching from irregular to continuous firing in VIP-INs

VIP-INs are strongly recruited by ascending cholinergic modulation in vivo (*Fu et al., 2014*), and M-current is characteristically inhibited by muscarinic receptor activation (*Brown and Passmore, 2009*). Therefore, we hypothesized that acetylcholine would activate IS VIP-INs and induce switching to a continuous firing mode. Bath application of the cholinomimetic carbachol (5 μM) produced a 10–15 mV depolarization in both CA and IS VIP-INs (*Figure 7A*). However, the suprathreshold firing properties of IS VIP-INs again converted to a 'CA-like' pattern, firing for a full 8 s depolarizing pulse with decreased irregularity. There was essentially no effect on the firing pattern of CA VIP-INs, nor was there any effect on the initial bursting characteristics of either IS or CA VIP-INs (*Figure 7B,C*). The effect of carbachol closely resembled the effect of linopirdine on IS VIP-INs. Similarly, we found that muscarinic, but not nicotinic stimulation, induced tonic firing in IS VIP-INs, while both types of cholinergic stimulation independently produced membrane potential depolarization of similar magnitude (4.24 ± 0.7 mV for muscarinic vs. 5.48 ± 1.4 mV for nicotinic; p>0.05; *Figure 7—figure supplement 1*). Hence, M-current inhibition via activation of muscarinic acetylcholine receptors represents a novel mechanism of neuromodulation in VIP-INs that is separate from the recently

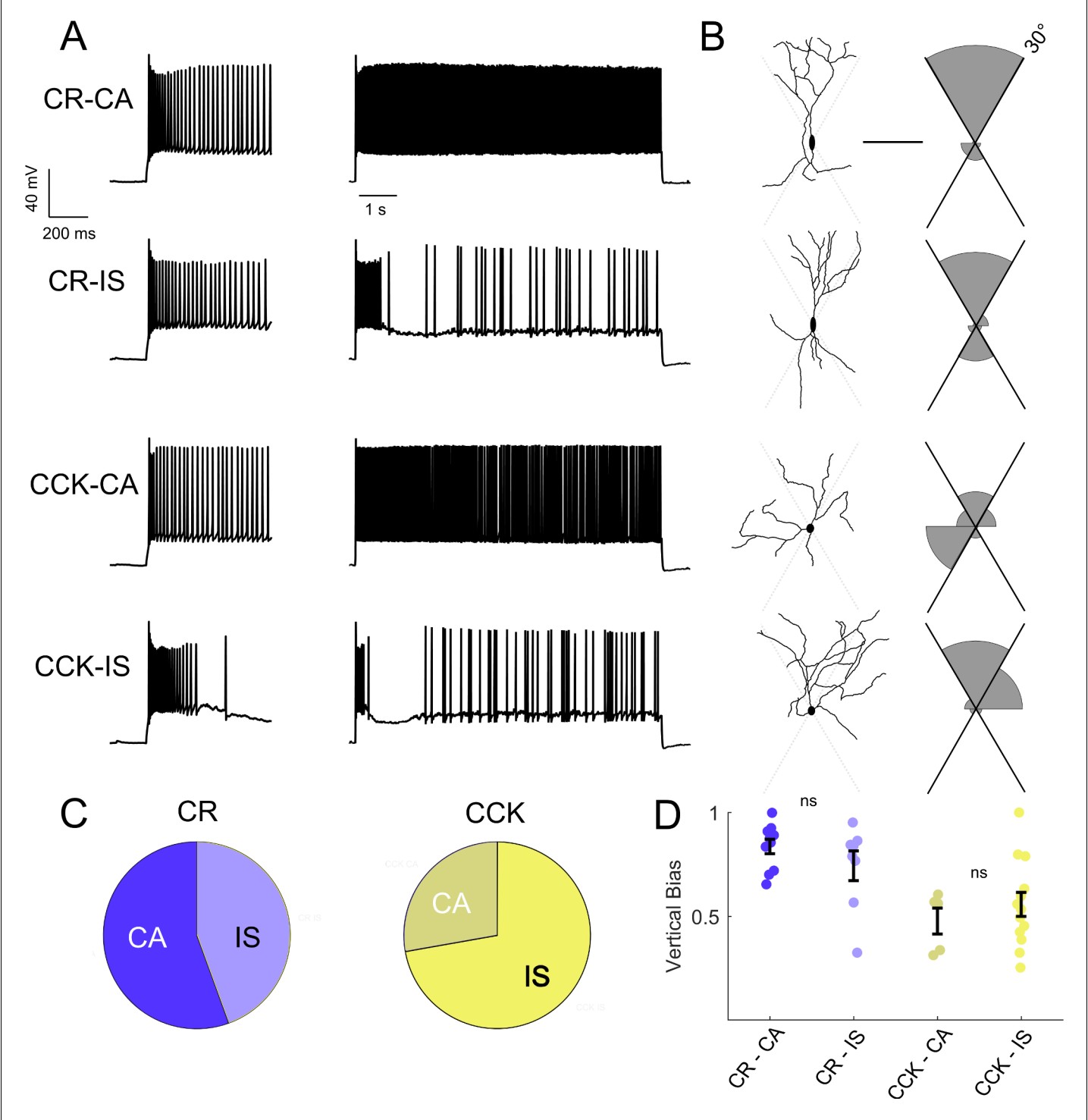

**Figure 5.** Intersectional expression of CR and CCK labels anatomical subsets of VIP-INs but does not correlate with IS vs. CA firing patterns. (A) Example traces from layer 2/3 VIP-INs in barrel cortex of adult CR-Cre/VIP-Flp and CCK-Cre/VIP-Flp mice. Both CA and IS firing patterns were observed in each intersectional population of VIP-INs. Insets on the left show the first 600 ms of each 8 s depolarization. (B) Morphological reconstruction of the proximal dendrites of the cells in A to illustrate dendritic orientation. Scale is 100 μm. The adjoining histogram quantifies the proportion of total dendrite that lies within or outside of 30° from vertical (perpendicular to the pial surface). Black axes indicate the length of histogram corresponding to 100%. (C) Proportion of CR and CCK VIP-INs that are CA vs. IS ($n$ = 17 total VIP-INs from 3 CR-Cre/VIP-Flp mice and $n$ = 19 VIP-INs from CCK-Cre/VIP-Flp mice); p>0.1 by Chi-square test. (D) Individual cells (*dots*) and population mean ± SEM for the vertical bias of each cell calculated from the morphological reconstructions in B. Vertical bias is the percent of dendrite within 30° of the line perpendicular to the pial surface, with a value

*Figure 5 continued on next page*

*Figure 5 continued*

of 1 corresponding to a perfectly bipolar shape. CA and IS VIP-INs in CR-Cre/VIP-Flp mice are mostly bipolar, with high vertical bias; CA and IS VIP-INs from CCK-Cre/VIP-Flp mice are multipolar, with vertical bias of ~ 0.5. Traditional Scholl analysis did not clearly illustrate this key difference between VIP-INs.

DOI: https://doi.org/10.7554/eLife.46846.016

described tonic depolarization caused by nicotinic acetycholine receptor activation (*Askew et al., 2019*).

## A single compartment model recapitulates the interaction between M-current and Na⁺ current density in the determination of VIP-IN discharge pattern

Taken together, evidence for a broad distribution of *Scn1a* across INs from RNA sequencing data, our own immunohistochemistry showing Nav1.1 protein on the axon of most VIP-INs, as well as the response of both IS and CA VIP-INs to Hm1a, suggests that Nav1.1 is likely present in both IS and CA VIP-INs, but that action potential generation in IS VIP-INs is more affected by heterozygous loss of Nav1.1. This could be due to a combination of lower non-Nav1.1-mediated Na⁺ current in IS VIP-INs (such that these cells are more reliant on Nav1.1 for spike generation and repetitive firing), combined with relatively higher expression of M-current mediated by Kcnq5-containing KCNQ channels. To further explore this question, we constructed a classic Hodgkin-Huxley (H-H) single compartment model, adding a slowly activating potassium current (gKS) to simulate the influence of M-like current (*Stiefel et al., 2013*) in VIP-INs (*Figure 8*).

After setting the standard H-H parameters to approximate the firing pattern of a typical WT CA VIP-IN (see Materials and methods), we increased M-current density to produce a response qualitatively similar to WT IS VIP-INs. Then, we performed a parameter sweep, varying both the M-current and Na⁺ current density to simulate the effect of loss of one copy of *Scn1a* in the presence or absence of M-current. Reducing Na⁺ current by up to 50% in models with little or no M-current had only minor effects on excitability (*Figure 8*), similar to our recordings from CA VIP-INs in *Scn1a*⁺/⁻ mice. However, in models with increasing amounts of M-current, reducing Na⁺ current density had a progressively more dramatic effect, leading to action potential failure after only a few action potentials (*Figure 8*), similar to what we observed experimentally in *Scn1a*⁺/⁻ IS VIP-INs. Therefore, both the CA and IS-VIP firing patterns that characterize VIP-IN diversity, as well as the effect of loss of one copy of *Scn1a*, are captured in a simple model with continuous distributions of both Na⁺ and M-current. This supports our conclusion that most or all VIP-INs express Nav1.1, and that IS VIP-INs – which have a prominent M-current mediated by Kcnq5-containing KCNQ channels – are selectively dysfunctional in *Scn1a*⁺/⁻ mice.

## Discussion

### VIP-INs express Nav1.1 and are hypoexcitable in *Scn1a*⁺/⁻ mice

VIP-INs exhibited impaired action potential generation in *Scn1a*⁺/⁻ mice consistent with decreased Na⁺ channel expression, and this deficit could be rescued with the Nav1.1-specific modulator Hm1a, supporting the presence of Nav1.1 subunit-containing Na⁺ channels in VIP-INs. We found that these deficits were localized to a large subset of VIP-INs that exhibited irregular spiking (IS). This basic finding was consistent across cortical areas, and, importantly, was seen at early developmental time points (P18-21) as well as in juvenile/young adult mice (P30-55). Interestingly, Hm1a had a larger effect on IS VIP-INs than on PV-INs, consistent with the observation that IS VIP-INs may be more profoundly impaired than PV-INs in *Scn1a*⁺/⁻ mice. This persistent VIP-IN dysfunction in *Scn1a*⁺/⁻ mice is distinct from the time course of PV-IN dysfunction, which we previously found is transient and delimited to an early developmental period when assessed via the same approach (*Favero et al., 2018*). Such a time course is consistent with the known natural history of Dravet syndrome in human patients: it is well established that seizure frequency typically decreases during early childhood, while cognitive deficits remain, with moderate to severe intellectual disability and features, or a formal diagnosis of, autism spectrum disorder (*Berkvens et al., 2015*; *Genton et al., 2011*; *Han et al.,*

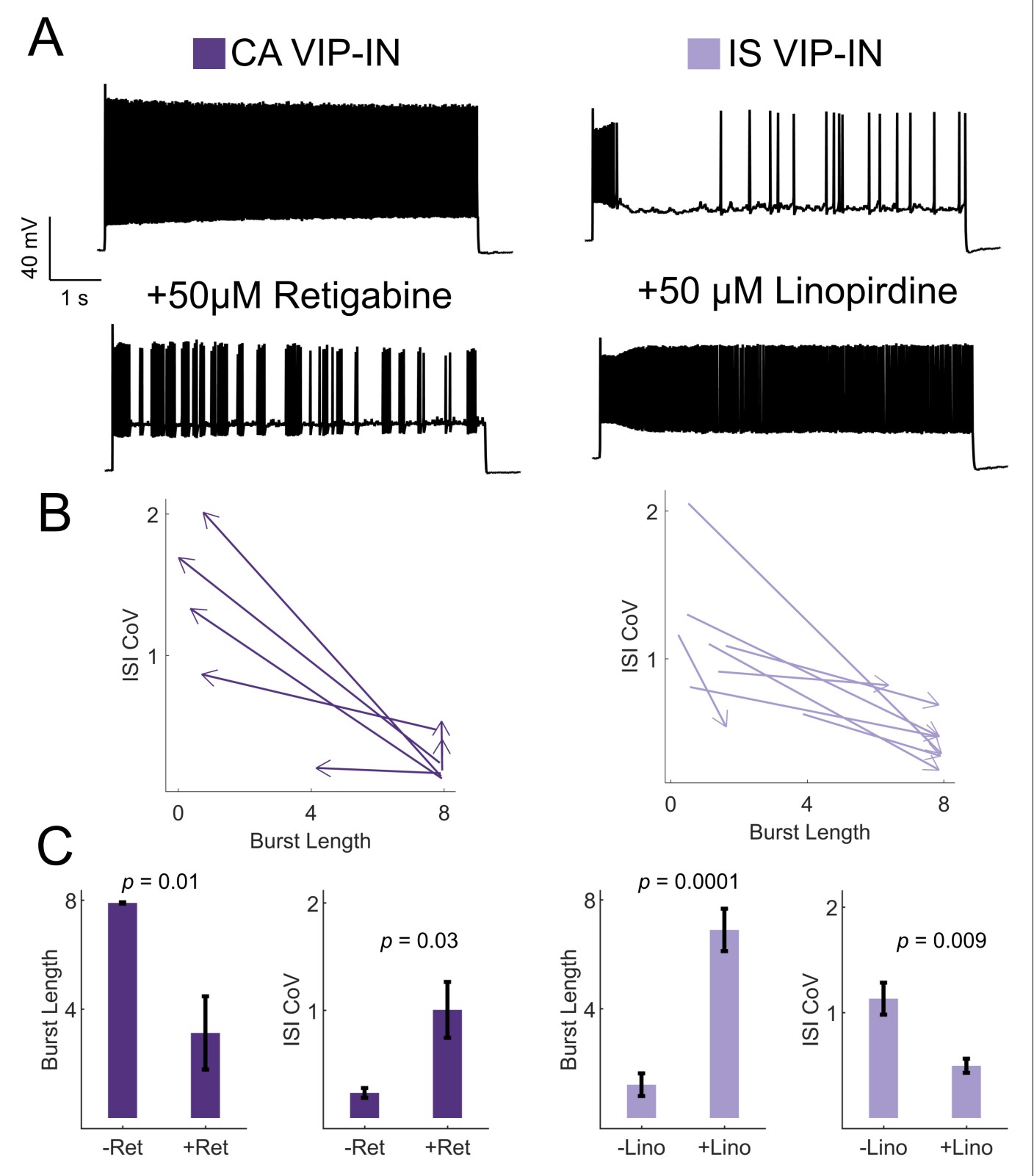

**Figure 6.** KCNQ channels regulate VIP-IN firing patterns. (**A**) Example traces from representative CA and IS VIP-INs in response to 8 s depolarization at 2X rheobase before and after bath application of either the KCNQ activator retigabine (*left*) or inhibitor linopirdine (*right*), respectively. (**B**) The response of *n* = 7 CA and *n* = 8 IS cells from 4 WT.VIP-Cre.tdT mice to the above drug application. Arrows indicate the direction of change in ISI CoV value and burst length (as in *Figure 2C*) after drug appliction, with the base of the arrow corresponding to the initial measurment, and the arrowhead
*Figure 6 continued on next page*

*Figure 6 continued*
corresponding to the values measured after drug application. (**C**) Mean ± SEM of each value in B before and after drug application, with *p*-values and significance determined by a paired students t-test.
DOI: https://doi.org/10.7554/eLife.46846.017
The following figure supplements are available for figure 6:

**Figure supplement 1.** Kcnq5 is selectively expressed in a subset of VIP-INs.
DOI: https://doi.org/10.7554/eLife.46846.018
**Figure supplement 2.** Kcnq5 but not Kcnq3 is expressed in VIP-INs.
DOI: https://doi.org/10.7554/eLife.46846.019
**Figure supplement 3.** Linopirdine has no effect on CA VIP-IN excitability.
DOI: https://doi.org/10.7554/eLife.46846.020

*2012*; *Li et al., 2011*). As VIP-INs are considered to have a role as disinhibitory elements in cortical circuits, it might seem counterintuitive that such cells might be dysfunctional in an epilepsy syndrome; however, it may be the case that early epilepsy in driven by transient PV-IN abnormalities while cognitive impairment persists in part due to ongoing VIP-IN hypofunction.

## Two distinct firing patterns of VIP-INs revealed by response to long depolarization

Previous studies have demonstrated a diversity of VIP-IN electrophysiological properties that do not clearly correlate with other molecular and anatomical markers of VIP-INs (*von Engelhardt et al., 2007*; *He et al., 2016*; *Kawaguchi and Kubota, 1997*; *Porter et al., 1998*; *Prönneke et al., 2015*). We also observed a seemingly diverse range of firing patterns in VIP-INs that did not clearly correlate with anatomy or proposed VIP-IN molecular markers CR and CCK (*Figure 5*). Using the response to longer (8–10 s) depolarizations, k-means clustering facilitated clear division into two electrophysiological groups, which we refer to as IS and CA VIP-INs (*Figure 2*; *Porter et al., 1999*), each of which comprise approximately 50% of VIP-INs in layer 2/3 in both barrel cortex and V1 at both P18-21 and P0-55. IS and CA firing patterns were robust to a variety of stimulation protocols, including a slow ramp current injection and prepulse step depolarization to inactivate T-type calcium currents that underlie burst firing in a subset of VIP-INs (*Prönneke et al., 2018*) (*Figure 2—figure supplement 1*). Therefore, irregular spiking is distinct from, yet partially overlaps with, bursting. However, our data clearly indicated that brief current pulses on the order of hundreds of milliseconds (such as are standard in slice physiology experiments) cannot establish whether a VIP-IN is IS or CA, as many IS VIP-INs fire continuously for 600 ms or more prior to transition to an IS pattern. This distinction may be relevant for ongoing efforts to create a comprehensive classification of VIP-INs (*Gouwens et al., 2019*; *He et al., 2016*; *Prönneke et al., 2015*; *Prönneke et al., 2018*). It will be important to investigate the physiological relevance of differences in firing patterns identified in acute brain slice experiments and how this relates to the in vivo activity of VIP-INs. While the stimuli used here are very different than naturalistic stimuli in the intact organism, such experiments reveal details related to the complement of ion channels expressed by VIP-INs as well as modulation by neurotransmitters that may be highly relevant in vivo.

## M-current mediated by Kcnq5-containing K⁺ channels regulates VIP-IN excitability and mediates response to cholinergic neuromodulation

We divided VIP-INs into IS and CA cells based on the presence of an initial period of regular firing followed by a suppression of firing and subsequent irregular spiking (as define by a high ISI coefficient of variation). This pattern suggested the expression of a slowly-activating potassium current such as M-current (*Brown and Passmore, 2009*). Transcriptomics data support our finding that most VIP-INs express Nav1.1 and suggests that a subset of VIP-INs express the KCNQ channel subunit Kcnq5 (*Paul et al., 2017*; *Tasic et al., 2016*), which we confirmed with immunohistochemistry (*Figure 6—figure supplement 2*). We hypothesized that M-current in IS VIP-INs drives this distinctive firing pattern, with CA VIP-INs express low levels of M-current that is insufficient to induce irregular firing. We were able to bidirectionally modulate VIP-IN firing patterns by pharmacologically blocking or enhancing M-current in IS and CA VIP-INs respectively, supporting this hypothesis. However, the

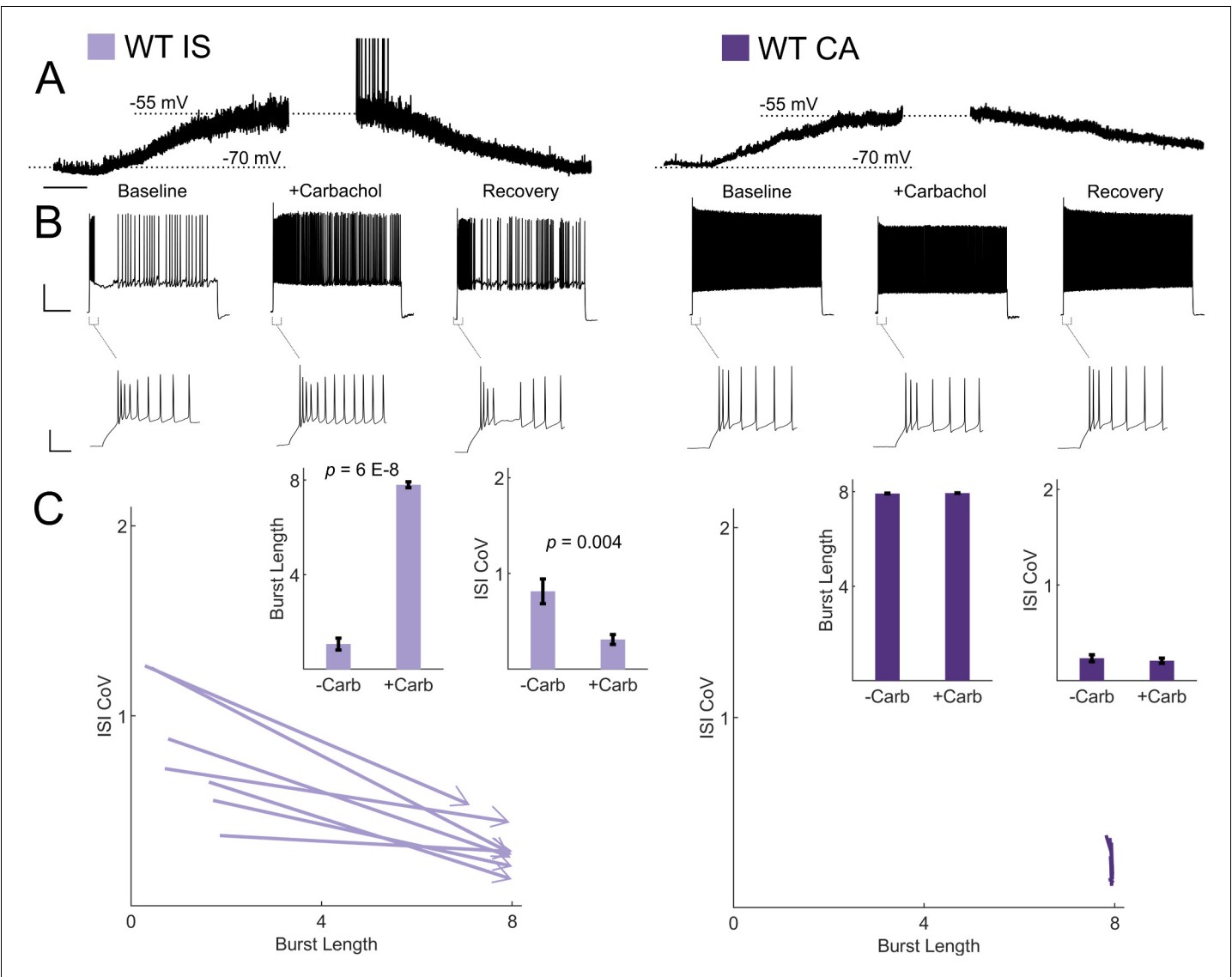

**Figure 7.** Cholinergic modulation induces switching from irregular to continuous firing in IS VIP-INs. (A) Depolarization in resting membrane potential of IS and CA VIP-INs with bath application of 5 µM Carbachol. Horizontal scale bar, 1 min. In addition to membrane depolarization, there was a notable increase in synaptic activity, and in some cases spontaneous firing. Population mean ± SEM of the change in membrane potential was 13 ± 5 and 10 ± 5 mV for IS and CA VIP-INs respectively (p>0.05). B)Action potential trains elicited with 8 s long square depolarizations at 2X rheobase current injections before, during, and after washout of 5 µM carbachol. In each case, the membrane potential was offset to −70 mV by direct current injection. Scale bars, 40 mV and 1 s. Insets showing no change in the initial bursting characteristics at suprathreshold current injections with the application of carbachol. Scale bars, 40 mV and 50 ms. C)Quantification of changes in firing patters of *n* = 7 IS and *n* = 8 CA VIP-INs from 3 WT.VIP-Cre.tdT mice in response to carbachol. All IS cells showed a qualitative switch to a continuous firing mode represented by increased burst length and decreased ISI CoV (indicated by arrowheads as in *Figure 6*); *p*-values determined by paired students' t-test. There were no apparent changes in CA VIP-IN firing patterns with carbachol application (arrowheads omitted for clarity).

DOI: https://doi.org/10.7554/eLife.46846.021

The following figure supplement is available for figure 7:

**Figure supplement 1.** Muscarinic but not nicotinic receptor activation is sufficient to induce irregular-tonic switching in IS VIP-INs.
DOI: https://doi.org/10.7554/eLife.46846.022

level of M-current in CA VIP-INs appears to be small enough such that blocking M-current in these cells has no effect on their firing properties (*Figure 6—figure supplement 3*). Even though our classification highlighted prominent differences in excitability, most or all VIP-INs do appear to express some level of both Nav1.1 and Kcnq2/5. It is possible that the electrophysiological properties of VIP-

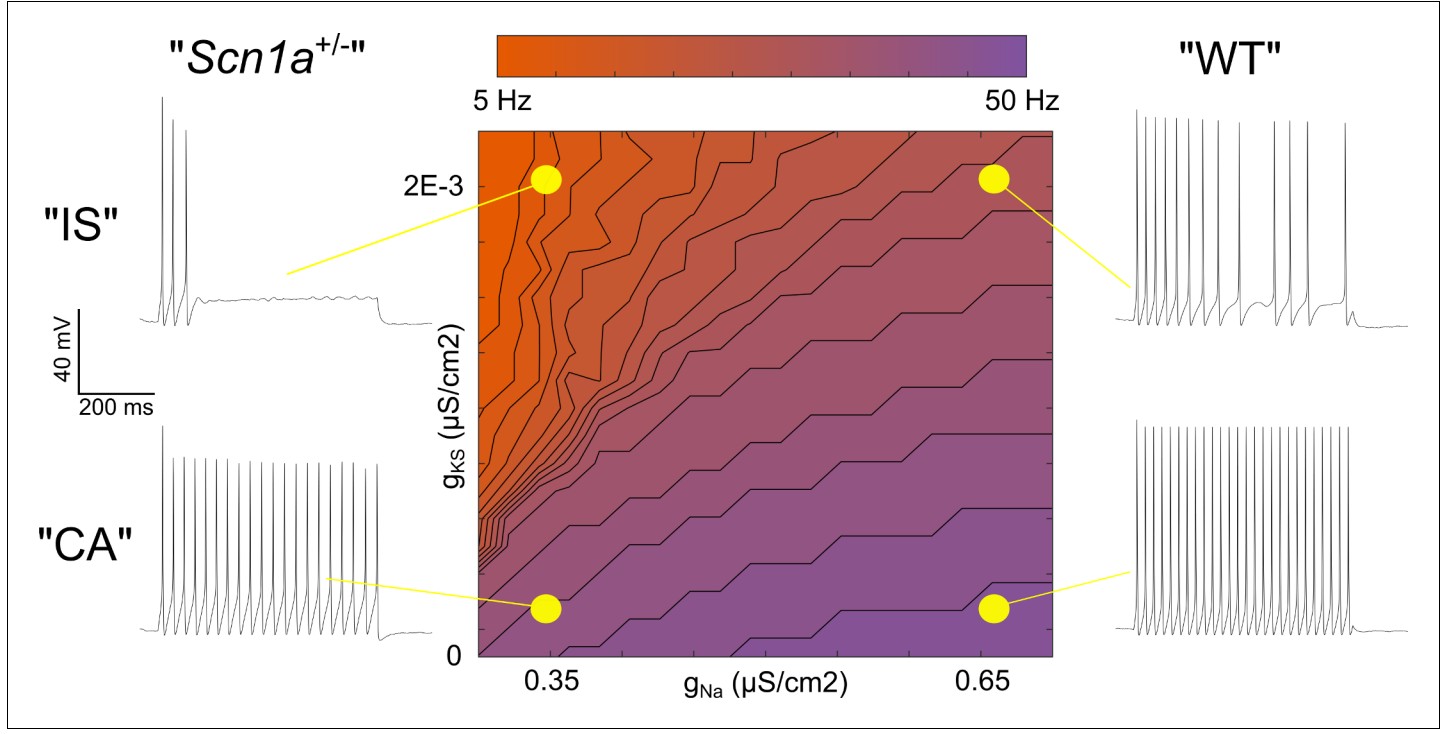

**Figure 8.** Single compartment model of a VIP-IN illustrates the interaction between M-current and Na$^+$ current. A single compartment Hodgkin-Huxley style conductance-based model using standard differential equations. Heat map indicates firing frequencies elicited with a 600 ms X 100 pA test pulse on a linear scale. The model was initially modified to approximate the intrinsic properties and firing rate of a typical WT CA VIP-IN observed experimentally. A slowly activating K$^+$ current with a fixed time constant (gKS) was then added to approximate the presence of M-current; in the presence of Gaussian distributed noise, this induced irregular spiking similar to that seen in IS VIP-INs (*top right*). Then, the amount of total Na$^+$ current density (gNa) in the model was varied. Models with low gNa and little or no gKS showed minimal impairment with a ~ 10–20% decrease in spike height and firing frequency. Models with low gNa and medium to high levels of gKS showed much more profound impairment, with complete collapse of repetitive action potential generation. The Matlab code used to generate this model is available in *Figure 8—source code 1* (model) and *Figure 8—source code 2* (figure generation).

DOI: https://doi.org/10.7554/eLife.46846.023

The following source code is available for figure 8:

**Source code 1.** Hodgkin-Huxley VIP-IN model Matlab code.
DOI: https://doi.org/10.7554/eLife.46846.024

**Source code 2.** Hodgkin-Huxley VIP-IN model code for Figure generation.
DOI: https://doi.org/10.7554/eLife.46846.025

INs defined here do not constitute clearly delimited 'subtypes' per se, but rather exist along a continuum of active properties which only partially overlap with molecular markers (such as CR and CCK) and morphology. Nevertheless, an irregular firing pattern that can be modulated by cholinergic suppression of M-current appears to be a durable feature of ~ 50% of VIP-INs.

If Kcnq5 and Nav1.1 in VIP-INs have overlapping distributions of expression, how does *Scn1a* haploinsufficiency lead to selective dysfunction of IS VIP-INs? To address this, we constructed a single compartment model with a voltage-gated Na$^+$ conductance and M-type K$^+$ conductance to simulate the influence of superimposed M-current on neuronal excitability in the context of varied Na$^+$ current density. This modeling recapitulated several important characteristics of our data, including the diversity of VIP-IN firing patterns, the role of M-current in this diversity, and the interaction between Na$^+$ current and M-current. The model illustrates how loss of Nav1.1 could have drastically different effects in IS vs. CA VIP-INs based on electrophysiological context, such as the overall level of M-current and the degree of reliance on Nav1.1.

VIP-INs are recruited during shifts in attentional state, and their activity is important for the cortical state transition that occurs in response to cholinergic neuromodulation (*Batista-Brito et al., 2017*; *Fu et al., 2014*; *Reimer et al., 2014*; *Walker et al., 2016*). Our data suggest that cholinergic

input may enhance VIP-IN excitability during cortical state transitions by inhibiting M-current. We found that, while application of carbachol depolarized both IS and CA VIP-INs, this manipulation selectively modulates IS VIP-IN excitability by inducing a switch from IS to tonic firing mode through muscarinic acetylcholine receptor activation. This effect may be important during shifts in attentional state, which are known to recruit VIP-IN activity. Our data also implies that IS VIP-IN dysfunction in $Scn1a^{+/-}$ mice may play a particularly important role in the activity of IS VIP-INs during such state changes. Overall, our findings identify a novel cellular locus of dysfunction in Dravet Syndrome which may be relevant to both the epilepsy and the severe cognitive impairments characteristic of this disease.

# Materials and methods

## Key resources table

| Reagent type (species) or resource | Designation | Source or reference | Identifiers | Additional information |
|---|---|---|---|---|
| Genetic reagent (*M. musculis*) | 129S-Scn1atm1Kea/Mmjax | Jax | RRID:MMRRC_037107-JAX | Dr. Jennifer A. Kearney, Northwestern University |
| Genetic reagent (*M. musculis*) | Viptm1(cre)Zjh/J | Jax | RRID:IMSR_JAX:010908 | |
| Genetic reagent (*M. musculis*) | B6;129P2-Pvalbtm1 (cre)Arbr/J | Jax | RRID:IMSR_JAX:008069 | |
| Genetic reagent (*M. musculis*) | B6J.Cg-Ssttm2.1 (cre)Zjh/MwarJ | Jax | RRID:IMSR_JAX:028864 | |
| Genetic reagent (*M. musculis*) | Rosa- CAG-LSL-td Tomato | Jax | RRID:IMSR_JAX:007914 | |
| Genetic reagent (*M. musculis*) | 129S6.SvEvTac | Taconic Biosciences | RRID:IMSR_TAC:129sve | |
| Genetic reagent (*M. musculis*) | C57BL/6J | Jax | RRID:IMSR_JAX:000664 | |
| Genetic reagent (*M. musculis*) | Ccktm1.1(cre)Zjh/J | Jax | RRID:IMSR_JAX:012706 | Drs. Bernardo Rudy and Robert Machold, NYU |
| Genetic reagent (*M. musculis*) | B6(Cg)-Calb2tm1 (cre)Zjh/J | Jax | RRID:IMSR_JAX:010774 | Drs. Bernardo Rudy and Robert Machold, NYU |
| Genetic reagent (*M. musculis*) | Viptm2.1(flpo)Zjh/J | Jax | RRID:IMSR_JAX:028578 | Drs. Bernardo Rudy and Robert Machold, NYU |
| Genetic reagent (*M. musculis*) | B6.Cg-Gt(ROSA) 26Sortm80.1 (CAGCOP4*L132C/ EYFP)Hze/J | Jax | RRID:IMSR_JAX:025109 | Drs. Bernardo Rudy and Robert Machold, NYU |
| Recombinant DNA reagent | AAV.CAG.Flex. tdTomato | Penn Vector Core | AV-9-ALL864 | 2XE + 13 GC/mL |
| Peptide, recombinant protein | Hm1a | Alomone | STH-601 | 1 µM / 50 nM |
| Chemical compound, drug | Biocytin conjugated to Alexa Fluor 488 | Invitrogen | A12924 | 0.5% |
| Chemical compound, drug | Linopirdine | Sigma | L134 | 50 µM |
| Chemical compound, drug | Retigabine | Alomone | D-23129 | 50 µM |
| Chemical compound, drug | Carbamoylcholine chloride | Sigma | C4382 | 5 µM |
| Chemical compound, drug | Muscarine | Sigma | M6532 | 5 µM |

*Continued on next page*

*Continued*

| Reagent type (species) or resource | Designation | Source or reference | Identifiers | Additional information |
|---|---|---|---|---|
| Chemical compound, drug | 4-Acetyl-1, 1-dimethylpiperazinium iodide | Tocris | 0352 | 1 µM |
| Software, algorithm | Pclamp 10 | Clampfit | RRID:SCR_011323 | V10.0 |
| Software, algorithm | Matlab | Mathworks | RRID:SCR_001622 | 2019a |
| Software, algorithm | Gramm | *Morel, 2018* | | Data visualization for Matlab |
| Antibody | Anti-Nav1.1 sodium channel, clone K74/71 | NeuroMab | RRID:AB_10671830 | 1:500, IF |
| Antibody | KCNQ5 Polyclonal Antibody | Invitrogen | RRID:AB_2736022 | 1:500, IF |
| Antibody | Anti-KCNQ3 Antibody | Alomone labs | RRID:AB_2040103 | 1:200, IF |
| Antibody | Goat anti-Mouse IgG1 Cross-Adsorbed Secondary Antibody, Alexa Fluor 488 | Invitrogen | RRID:AB_2535764 | 1:1000, IF |
| Antibody | Goat anti-Rabbit IgG (H + L) Cross-Adsorbed Secondary Antibody, Alexa Fluor 488 | Invitrogen | RRID:AB_143165 | 1:1000, IF |

## Experimental animals

All procedures and experiments were approved by the Institutional Animal Care and Use Committee at the Children's Hospital of Philadelphia and were conducted in accordance with the ethical guidelines of the National Institutes of Health. Both male and female mice were used in equal proportions. After weaning at P21, mice were group-housed with up to five mice per cage and maintained on a 12 hr light/dark cycle with *ad libitum* access to food and water.

Mouse strains used in this study included: $Scn1a^{+/-}$ mice on a 129S6.SvEvTac background (RRID: MMRRC_037107-JAX) generated by a targeted deletion of exon 1 of the *Scn1a* gene, VIP-Cre mice (Viptm1(cre)Zjh/J; RRID:IMSR_JAX:010908 on a mixed C57BL/6;129S4 background), PV-Cre mice (B6;129P2-Pvalbtm1(cre)Arbr/J; RRID:IMSR_JAX:008069), SST-Cre mice (B6J.Cg-Ssttm2.1(cre)Zjh/MwarJ; RRID:IMSR_JAX: 028864), tdTomato reporter/Ai14 mice (Rosa- CAG-LSL-tdTomato; RRID: IMSR_JAX:007914; on a C57BL/6J background), wild-type 129S6.SvEvTac (Taconic Biosciences model #129SVE; RRID:IMSR_TAC: 129sve), and wild-type C57BL/6J (RRID:IMSR_JAX:000664).

Homozygous VIP-Cre mice were crossed to homozygous Ai14 mice to generate VIP-Cre.tdT double heterozygotes on a predominantly C57BL/6J background. Female VIP-Cre.tdT double heterozygotes were then crossed to male $129S6.Scn1a^{+/-}$ mice to generate Scn1a.VIP-Cre.tdT mice and WT. VIP-Cre.tdT littermate controls. The genotype of all mice was determined via PCR of tail snips obtained at P7 and was re-confirmed for each mouse after they were sacrificed for slice preparation. The same breeding scheme was used to generate Scn1a.PV-Cre.tdT (as in *Favero et al., 2018*). All mice used for experiments were on a near 50:50 129S6:B6J background, and $Scn1a^{+/-}$ mice on this background have been shown to replicate the core phenotype of Dravet Syndrome (*Miller et al., 2014*; *Mistry et al., 2014*). We observed similar rates of spontaneous death (17/52, or 32%, of Scn1a.VIP-Cre.tdT mice not used for experiments by P55; *Favero et al., 2018*) and were able to thermally induce seizures in 100% (9/9) of P21-50 Scn1a.VIP-Cre.tdT mice at or below 42°C core body temperature, supporting the validity of using these mice as a tool to study VIP-IN function in Dravet Syndrome.

We used an intersectional genetic approach (*He et al., 2016*; *Taniguchi et al., 2011*) to target CCK and CR expressing VIP-INs. These mice were a cross between either CCK-Cre (Ccktm1.1(cre) Zjh/J; RRID: IMSR_JAX:012706) or CR-Cre (B6(Cg)-Calb2tm1(cre)Zjh/J; RRID: IMSR_JAX:010774)

crossed to a VIP-Flp (Viptm2.1(flpo)Zjh/J; RRID: IMSR_JAX:028578) and fluorescent reporter Ai80 (B6.Cg-Gt(ROSA)26Sortm80.1(CAG-COP4*L132C/EYFP)Hze/J; RRID: IMSR_JAX:025109).

## Stereotaxic injections

In a subset of experiments at the P30-50 age, VIP-INs from double transgenic *Scn1a*.VIP-Cre and WT.VIP-Cre littermates from the cross described above were labeled via stereotaxic injection of AAV.CAG.Flex.tdTomato (Penn Vector Core, AV-9-ALL864) at approximately P25. Briefly, P25 mice were anesthetized with isoflurane (induction, 3–4%; maintenance, 1–1.5%) and body temperature and breathing were continuously monitored. A small craniotomy approximately 1 mm posterior and 3 mm lateral to bregma was made to allow insertion of a 50–75 µm tip diameter glass pipette driven by a Nanoject III (Drummond Scientific). 100 nL of 2XE + 13 GC/mL of AAV9 diluted in sterile PBS was injected at 20 nL/min. The pipette was held in place for 10 min to allow the virus to spread, and was then slowly removed; the scalp was sutured closed, and the mouse allowed to recover.

## Acute slice preparation

Mice were anesthetized with isoflurane and transcardially perfused with ice cold artificial cerebral spinal fluid (ACSF) containing (in mM): NaCl, 87; sucrose, 75; KCl, 2.5; $CaCl_2$, 1.0; $MgSO_4$, 6.0; $NaHCO_3$, 26; $NaH_2PO_4$, 1.25; glucose, 10, and equilibrated with 95% $O_2$ and 5% $CO_2$. The brain was removed to cold ACSF, then mounted on a holder of the Leica VT-1200S vibratome and sliced at 300–350 µm thickness. Slices were allowed to recover for 30 min in ACSF warmed to 30°C, then maintained at room temperature for up to 6 hr before recording. Slices were transferred to a recording chamber on the stage of a BX-61 upright microscope and continuously perfused with recording solution at 30–32°C and 3 mL/min that contained, in mM: NaCl, 125; KCl, 2.5; $CaCl_2$, 2.0; $MgSO_4$, 1.0; $NaHCO_3$, 26; $NaH_2PO_4$, 1.25; glucose, 10.

## Slice recordings

VIP-INs and PV-INs were identified by tdT expression visualized with epifluorescence. Pyramidal cells were identified by morphology under infrared differential interference contrast (IR-DIC) and the presence of a regular-spiking firing pattern. Whole-cell recordings were obtained from superficial (layer 2/3) primary somatosensory cortex (S1; 'barrel') and visual cortex (V1) with between 1–4 (usually 1 or 2) cells recorded from each slice in either single or paired configuration. Patch pipettes were pulled from borosilicate glass using a P-97 puller (Sutter Instruments) and filled with intracellular solution containing (in mM): K-gluconate, 130; KCl, 6.3; EGTA, 0.5; $MgCl_2$, 1.0; HEPES, 10; Mg-ATP, 4.0; Na-GTP, 0.3; pH was adjusted to 7.30 with KOH, and osmolarity adjusted to 285 mOsm with 30% sucrose. Where indicated, intracellular solution also contained 0.5% biocytin conjugated to Alexa Fluor 488 (Invitrogen) for 2P imaging. Pipettes had a resistance of 4–6 MΩ when filled and placed in recording solution.

Voltage was sampled at 50 kHz with a MultiClamp 700B amplifier (Molecular Devices), filtered at 10 kHz, digitized using a DigiData 1550A, and acquired using pClamp10 software. Recordings were discarded if the cell had an unstable resting membrane potential and/or a membrane potential greater than −50 mV, or if access resistance increased by > 20% during the recording. We did not correct for liquid junction potential.

## Electrophysiology data analysis

All analysis was performed blind to genotype using Matlab (Mathworks) with quality control using manual confirmation in Clampfit (pCLAMP). Resting membrane potential ($V_m$) was calculated using the average value of a 1 s sweep with no direct current injection. Input resistance ($R_m$) was calculated using the average response to small hyperpolarizing current injections near rest using $R_m = \Delta V/I$ for each sweep. AP threshold was calculated as the value at which the derivative of the voltage (dV/dt) first reached 10 mV/ms. Spike height refers to the absolute maximum voltage value of an individual AP, while spike amplitude was calculated as the difference between spike height and AP threshold for a given AP. AP rise time is the time from AP threshold to the peak of the AP. AP half-width (AP 1/2-width) is defined as the width of the AP (in ms) at half-maximal amplitude (half the voltage difference between the AP threshold and peak). AP afterhyperpolarization (AHP) amplitude is calculated

as the depth of the afterhyperpolarization (in mV) relative to AP threshold. Unless indicated, all quantification of single spike properties was done using the first AP elicited at rheobase (below).

Rheobase was determined as the minimum current injection that elicited APs using a 600 ms sweep at 10 pA intervals. Maximal instantaneous firing was calculated using the smallest interspike interval (ISI) elicited at near-maximal current injection. Maximal steady-state firing was defined as the maximal mean firing frequency during the last 300 ms of a suprathreshold 600 ms current injection, with a minimum requirement for a spike being an amplitude of 40 mV with a clear AP threshold (above) and height overshooting at least $-10$ mV. We found the combination of these two measures to best describe VIP-IN firing rates, as some cells fired a very brief burst of APs at high frequency, while others fired continuously with little or no initial bursting. All *I*-f plots were created using the steady-state firing calculated for each current step, counting failures as 0 for subsequent current steps. Population raster plots were constructed by taking a 20 ms sliding average of the instantaneous firing rate for each current step and averaged over each group of cells.

## Firing pattern classification

We used a longer depolarizing current injection (generally, 8–10 s) which was found to highlight features of VIP-IN firing that were not apparent with briefer, 600 ms current injections. The coefficient of variation of the ISI (ISI CoV) was used to quantify irregularity of repetitive AP discharge, and was defined as the standard deviation divided by the mean of all ISI's from a single 8 s or 600 ms sweep at 2-times rheobase current injection. The burst length of a cell was defined by taking the time of the last spike occurs prior to an abrupt cessation of firing lasting > 150 ms. This value was determined in an unbiased way by taking the mean + 2 times standard deviation of all ISI values from our dataset. We used the same cutoff to calculate an 'apparent' burst length when using data from 600 ms sweeps. We used k-means clustering with *n* = 2 groups to cluster VIP-INs from both WT and $Scn1a^{+/-}$ mice using the variables of ISI CoV and burst length from 8 s current injections at 2-times rheobase. Clustering was validated with > 95% agreement with blind manual classification when dividing cells based on the provided description of continuous firing with spike frequency adaptation ('CA') versus initial burst followed by irregular spiking ('IS'). It was not possible to use 600 ms sweeps alone to replicate these two groups, either with k-means clustering of electrophysiological properties or manual classification.

## Voltage clamp recordings of acutely dissociated cells

We performed voltage clamp recordings from acutely dissociated cells prepared from neocortex of P18 WT.VIP-Cre.tdT mice. Briefly, the mouse brain was extracted into ice cold ACSF as described above. Then, 400 μm slices were cut on a vibratome, and the neocortex was manually separated from the underlying white matter using a scalpel. All neocortical sections were simultaneously place in dissociation media (Earl's balanced salt solution (EBSS)) with (in mM): NaCl, 117; KCl, 5.0; $NaHCO_3$, 26; $NaH_2PO_4$, 1.0; $CaCl_2$, 1.0; $MgSO_4$, 4.0; HEPES, 20; glucose, 10, equilibrated with 95% $O_2$:5% $CO_2$, and pH adjusted to 7.4 with NaOH. This was supplemented with 0.1% trypsin (Sigma), 1 mg/mL collagenase (Sigma), 100 U/mL DNAse1 (Sigma), and incubated for 10 min at 35° C. Dissociation was quenched by pouring the contents into a large volume of room temperature ACSF, and the neocortical sections were maintained for up to 4 hr. Prior to recording, individual sections were manually dissociated by triturating with a series of fire-polished glass pipettes decreasing in size, then immediately plated on a glass coverslip coated in poly-D lysine (Sigma). Cells were allowed to settle and adhere for 10 min, then the ACSF was exchanged for recording solution containing (in mM): NaCl, 135; KCl, 4; $CaCl_2$, 2; $MgCl_2$, 2; HEPES, 10; Glucose, 10; TEA-Cl, 10; $CdCl_2$, 0.1; with pH adjusted to 7.4 with NaOH. VIP-INs were identified with epifluorescence. Cells were recorded at room temperature (21–24° C) with patch pipettes with a resistance measured at 2.5–3.0 MΩ containing (in mM): NaF, 10 CsF, 110; CsCl, 20; EGTA, 2; HEPES, 10; with pH adjusted to 7.4 with KOH. $Na^+$ currents were recorded in cells that were stable and had an access resistance < 10 MΩ for 5 min after break in using a series of 100 ms voltage steps from $-80$ to 50 mV. Peak current was calculated as the max absolute value of the current response. Persistent current was calculated as a percentage of transient current using the mean current during the period between 80 and 100 ms after the initial voltage step. The time constant of inactivation (tau) was calculated by fitting a double

exponential to the decay of the Na$^+$ current, reporting the value for the dominant term. All values in *Figure 4* were calculated using the current elicited at 0 mV.

## Immunohistochemistry

To facilitate staining of Nav1.1 at the AES, we used very mild fixation (1% paraformaldehyde with 0.5% MeOH in PBS) described previously (*Alshammari et al., 2016*). Briefly, isoflurane-anesthetized mice were transcardially perfused; brains were removed and post-fixed in perfusate at RT for 1 hr. We then immediately cut 50 μm sections on a Leica VT-1200S vibratome, and then blocked and permeabilized the slices with 0.5% Triton X-100 (Sigma) and 10% normal goat serum in PBS for one hour at RT. We stained overnight at 4° C with a primary antibody directed against Nav1.1 (Neuro-Mab K74/71) in PBS with 3% bovine serum albumin (BSA, Sigma) and 0.5% Triton X-100. The following day, the slices were washed with PBS and stained with a secondary antibody, Alexa Fluor 488-conjugated goat anti-mouse IgG1 (Molecular Probes), in PBS, with 3% BSA and 0.5% Triton X-100. After washing, slides were cover-slipped and sealed before imaging on a Leica TCS SP8 confocal microscope. We examined VIP-INs under 80X magnification and detected Nav1.1 signal on very fine (0.5 μm) processes emanating from either the cell body or proximal dendrite. We used a cutoff of mean + 2 times the standard deviation of the Nav1.1 signal measured using repeated line scans on at least 5 μm (consecutive) of tdT-positive VIP-IN processes to define a 'positive' VIP-IN axon. For verification of Nav1.1 antibody specificity, we generated several *Scn1a*$^{-/-}$ mice by breeding two *Scn1a*$^{+/-}$ mice. For Kcnq3/5 staining, we used standard 4% paraformaldehyde, then performed staining as described above.

## Allen brain data access

Data were accessed directly from: 2015 Allen Institute for Brain Science. Allen Cell Types Database. Available from: celltypes.brain-map.org/rnaseq. This dataset represents over 20,000 cells pooled from primary mouse visual cortex (V1) and anterior lateral motor cortex (ALM). The provided RNA-Seq Data Navigator was used to restrict data selection to pyramidal cells and VIP-INs from layer 2/3 to match our experimental data. SST and PV-INs from all layers were also included as a reference. *ScnXa*, *KcnqX*, *Cck*, and *Calb2* expression levels were downloaded in February 2019 from the indicated cell types for offline analysis and plotting using Matlab.

## Slice pharmacology

We used several pharmacological agents including Hm1a (Alomone Labs STH-601), linopirdine (Sigma L134), retigabine (Alomone D-23129), the cholinomimetic carbamoylcholine chloride (carbachol; Sigma C4382), muscarine (Sigma M6532), and the nicotinic agonist 4-Acetyl-1,1-dimethylpiperazinium iodide (Tocris 0352). Hm1a, muscarine, and 4-Acetyl-1,1-dimethylpiperazinium iodide were dissolved in deionized water, aliquoted, and frozen at −20° C. Carbachol was stored as a powder at RT, and both linopirdine and Retigabine were stored as 100 mM stock solutions in DMSO (Sigma) aliquoted in −20°C. For pharmacological experiments in which drug was dissolved in DMSO, the same concentration of DMSO was present in control external solution. After a baseline recording, drugs were perfused in at 3 mL/min while continuously recording membrane potential; repeat measurements were performed 10 min after wash-in. A subset of cells were recorded for up to 1 hr following washout of each drug, as needed in some cases to observe complete or near-complete washout. For experiments using carbachol, repeat measurements were made 5 min after wash-in.

## 2 P imaging and morphological reconstruction

Where indicated, internal solution contained 0.5% biocytin conjugated to Alexa Fluor 488, or, in some cases, 50 mM Alexa Fluor alone. After obtaining the whole cell configuration, the cell was dialyzed for at least 20 min; then, the pipette was slowly removed to allow for the cell to reseal. Two-photon (2P) imaging was performed using a customized Bruker 2P microscope system with a MaiTai DeepSee Ti:Sapphire pulsed infrared laser (SpectraPhysics) directed through a modified Olympus BX-61 base equipped with a GaAsP photodetector (Hamamatsu). 3D image stacks were reconstructed using the built in ImageJ plugin Simple Neurite Tracer and compressed to a 2D trace which was analyzed using Matlab. We used 'vertical bias' as a summary statistic to capture the difference between bipolar and multipolar morphologies. We quantified the total length of dendrite that fell

within 30° of vertical (i.e., a line perpendicular to the pial surface), and then normalized to the total length of dendrite. Using this method, bipolar cells had a vertical bias close to 1, while multipolar cells with processes extending in all directions had vertical bias ranging from near-zero to close to 0.5.

## Hodgkin-Huxley VIP-IN model

All simulations were done using custom Matlab code available in *Figure 8—source code 1*. We used classic Hodgkin-Huxley equations with the single addition of a slow K$^+$ conductance, $g_{KS}$, as in *Stiefel et al. (2013)*. Here, we are using the additional slowly activating $g_{KS}$ to model in a generic fashion the effects of M-current mediated irregular spiking in VIP-INs. The equations for our model are

$$\frac{c_m dV}{dt} = g_{leak}(E_l - V) + m^3 h\, g_{Na}(E_{Na} - V) + n^3\, g_{KDR}(E_K - V) + s\, g_{KS}(E_K - V) + I_{inj} + I_{noise}$$

where *V* is the membrane voltage, *m* and *h* are the activation and inactivation states of the voltage gated Na$^+$ conductance, *n* is the activation variable of the delayed-rectifier K$^+$ conductance, and *s* is the activation variable of the slow K$^+$ conductance. These variables are determined by the set of differential equations:

$$\frac{dm}{dt} = (m_{inf} - m)/\tau_m$$

$$\tau_m = 1/(\alpha + \beta)$$

$$m_{inf} = \alpha/(\alpha + \beta)$$

The equations for *h*, *n*, and *s* all have the same form. For *m*, the expressions for α and β are given by

$$\alpha = 0.1(V + 38)/(1 - \exp(-(V + 38)/10)$$

$$\beta = 4\exp(-(V + 65)/17)$$

For *h*, the expressions for α and β are given by

$$\alpha = 0.05\exp(-(V + 55)/20)$$

$$\beta = 1/((\exp(-(V + 35)/10)) + 1)$$

For *n*, the expressions for α and β are given by

$$\alpha = 0.01(V + 55)/(1 - \exp(-(V + 55)/10)$$

$$\beta = 0.125\exp(-(V + 65)/80)$$

The expressions for *s* have a fixed $\tau_s$ = 300 ms and

$$s_{inf} = 1/(1 + \exp(-(V + 35)/5))$$

We manually fit the parameters of this model to resemble some of the key characteristics of VIP-INs, including a high input resistance (small $g_{leak}$) and small size (small $c_m$), as well as the approximate *I*-f curves generated by simulated current injections ($I_{inj}$). The conductances used were $g_{leak} = 0.03\,mS \cdot cm^{-2}$, $g_{KDR} = 5\,mS \cdot cm^{-2}$, $g_{Na} = 65\,mS \cdot cm^{-2}$, $g_{KS} = 2\,mS \cdot cm^{-2}$ where we varied $g_{Na}$ and $g_{KS}$ as described in *Figure 8*. The reversal potentials were $E_{Cl}$ = -70, $E_K$ = -70, and $E_{Na}$ = 50. Numerical integration was performed using the forward Euler method. We injected $I_{noise}$ throughout the simulation as a Gaussian white distribution with a max amplitude of 10 pA.

## Statistics and experimental design

Data from 62 total mice were used in this study for comparing intrinsic properties between WT and $Scn1a^{+/-}$ mice. A total of $n$ = 57 VIP-INs from P18-21 and $n$ = 86 VIP-INs from P30-55 WT mice, and $n$ = 41 VIP-INs from P18-21 and $n$ = 61 VIP-INs from P30-55 $Scn1a^{+/-}$ mice were included. We validated our core findings in a subset of experiments in primary visual cortex, with a total of $n$ = 28 VIP-INs from 4 WT.VIP-Cre.tdT mice and $n$ = 28 VIP-INs from 6 $Scn1a$.VIP-Cre.tdT mice age P18-21. All data were tested for normal distribution by performing a Shapiro-Wilk test and variances were estimated using Levene's test. Generally, intrinsic properties from WT and Scn1a VIP-INs had equal variances but occasional non-normal distributions. Thus, electrophysiological data were compared with Mann-Whitney U test for comparisons between two groups and Kruskal-Wallis test with a post hoc Dunn's test when comparing more than two groups. We observed no statistical difference between intrinsic properties of P18-21 and P30-55 WT VIP-INs and thus did not incorporate a model to account for changes in development (as in *Favero et al., 2018*). Therefore, we combined ages for most of our analysis. Between 3–10 cells were recorded per mouse. For most comparisons, each cell as treated as an $n$, but we confirmed all main findings via reanalysis with each animal as an $n$ (**Figure 3—figure supplement 1**). For active properties that involved repeated measures, such as current/frequency plots and spike rundown with repeated APs, we used a multivariate ANOVA to compare between two groups, with post hoc Bonferroni correction to estimate $p$ values for each measurement. For pharmacology experiments, a paired student's t-test was used to test for the effect of a drug within a single group. All statistical analysis was performed using built in Matlab functions. Data were visualized using built in Matlab functions, or using the gramm suite (*Morel, 2018*).

## Acknowledgements

This work was funded by NIH research grant K08 NS097633 and the Burroughs Wellcome Fund Career Award for Medical Scientists to EMG. We thank Dr. Jennifer A Kearney at Northwestern University for the generous gift of $Scn1a^{+/-}$ mice and Drs. Bernardo Rudy and Robert Machold at NYU for the gift of the VIP/CCK and VIP/CR mice. We thank Dr. Xiaohong Zhang for expert technical assistance.

# Additional information

### Funding

| Funder | Grant reference number | Author |
| --- | --- | --- |
| National Institute of Neurological Disorders and Stroke | K08 NS097633 | Ethan Michael Goldberg |
| Burroughs Wellcome Fund | Career Award for Medical Scientists | Ethan Michael Goldberg |
| National Institute of Neurological Disorders and Stroke | R01 NS110869 | Ethan Michael Goldberg |

The funders had no role in study design, data collection and interpretation, or the decision to submit the work for publication.

### Author contributions

Kevin M Goff, Data curation, Formal analysis, Investigation, Writing—original draft, Writing—review and editing; Ethan M Goldberg, Conceptualization, Resources, Formal analysis, Supervision, Funding acquisition, Investigation, Methodology, Writing—original draft, Project administration, Writing—review and editing

### Author ORCIDs

Kevin M Goff (ID) https://orcid.org/0000-0001-5862-0219
Ethan M Goldberg (ID) https://orcid.org/0000-0002-7404-735X

## Ethics

Animal experimentation: This study was performed in strict accordance with the recommendations in the Guide for the Care and Use of Laboratory Animals of the National Institutes of Health. All of the animals were handled according to approved institutional animal care and use committee (IACUC) protocol (#1152) of The Children's Hospital of Philadelphia. The protocol was approved by the IACUC Committee of The Children's Hospital of Philadelphia. All surgery was performed under isoflurane anesthesia, and every effort was made to minimize suffering.

## Decision letter and Author response

Decision letter https://doi.org/10.7554/eLife.46846.030
Author response https://doi.org/10.7554/eLife.46846.031

## Additional files

### Supplementary files
• Transparent reporting form
DOI: https://doi.org/10.7554/eLife.46846.026

### Data availability

All data generated or analysed during this study are included in the manuscript and supporting files. Source data files have been provided for Figure 3, Table 1 and Figure 8.

The following previously published dataset was used:

| Author(s) | Year | Dataset title | Dataset URL | Database and Identifier |
|---|---|---|---|---|
| Allen Institute for Brain Science | 2015 | Allen Cell Types Database | http://celltypes.brain-map.org/rnaseq | Allen Brain Institute Data Portal, rnaseq |

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
