## [Decision Letter]

Thank you for submitting your article "Vasoactive intestinal peptide-expressing interneurons are impaired in a mouse model of Dravet Syndrome" for consideration by *eLife*. Your article has been reviewed by three peer reviewers, and the evaluation has been overseen by a Reviewing Editor and Gary Westbrook as the Senior Editor. The following individuals involved in review of your submission have agreed to reveal their identity: Bernardo Rudy (Reviewer #1) and Bruce P Bean (Reviewer #3). The reviewers have discussed the reviews with one another and the Reviewing Editor has drafted this decision to help you prepare a revised submission.

Summary:

This study provides high quality data supporting new ideas regarding the pathogenesis of Dravet Syndrome (DS), a severe neurodevelopmental disorder caused by loss of function in the gene (SCN1A) encoding the Nav1.1 Na^+^ channel. The study provides strong functional evidence that, contrary to the prevalent assumption that only PV and SST GABAergic interneurons express Nav1.1, another interneuron class, VIP-expressing INs, also express this channel in their axons such that VIP IN firing is impaired. Although the effects of the reduction in Nav1.1 channels on the excitability of PV INs were transient, the hypoexcitability of VIP INs is maintained into adulthood. The authors suggest the intriguing hypothesis that the transient malfunction of PV neurons may contribute to the transient epileptic phenotype of Dravet syndrome, whereas the effect on VIP neurons may underlie the longer lasting cognitive impairment. Interesting new data are also provided on heterogeneity of VIP INs.

Essential revisions:

1) The data showing expression of Scn1a by VIP-INs are quite weak. This is important, as the authors note previous work citing limited expression of Nav1.1 in VIP-INs, and the conclusions largely rest on the idea that altered cell physiology is due to cell-autonomous loss of sodium channels in VIP-INs. The immunostaining has no negative control (is labeling lost/reduced in knockout mice?). Moreover, the labeling in Figure 1 looks completely non-specific. Accordingly, either the reagents should be validated, for example by examining staining in knockout mice, or the immunohistochemical results should be dropped from the study.

2) The authors present experiments showing that the peptide toxin Hm1a, which destabilizes inactivation of Nav1.1 channels, enhances firing of IS-type VIP neurons while having less dramatic effects on firing of CA type VIP neurons and on PV cells, and with no effect on firing of pyramidal neurons. These experiments are very nice. However, it was very strange that the authors make no mention of the paper by Richards et al. (PNAS 2018,115(34):E8077-E8085) with very similar experiments with Hm1a on a different mouse model of Dravet syndrome, showing enhancement of firing of hippocampal GABAergic neurons but no effect on firing of hippocampal pyramidal neurons, and also in vivo experiments showing efficacy of the toxin to reduce seizures and mortality. The Richards paper also presents a more detailed analysis of the effects of Hm1a on the gating of Nav1.1 channels than in the Osteen papers, as well as more comprehensive data on selectivity for Nav1.1 over other Na channels and K channels. The original Osteen et al. characterization of Hm1a toxin shows some effect on Nav1.2 as well, leaving the conclusion of Nav1.1 expression less clear. In addition, the dosing used in this study appears to be much too high. In the Richards paper they showed that Nav1.1 is sensitive to 1 nM toxin, and they used 5 nM in their slice experiments. Here, Goff and Goldberg used 1 μM, which may well not be specific. Since this experiments is important for demonstrating that the Na current in VIP neurons is actually from Nav1.1, I then this result should be supported with an Hm1a concentration of ~5 nM.

3) While the recordings are of good quality and many of the basic findings are reasonably well-supported, the overall presentation is confusing as to study goals. Is the focus about different classes of VIP-INs or about dysfunction in Dravet syndrome? This duality ultimately weakens the strength of conclusions in either case. The authors should address this issue in their revision.

---

## [Author Response]

Essential revisions:1) The data showing expression of Scn1a by VIP-INs are quite weak. This is important, as the authors note previous work citing limited expression of Nav1.1 in VIP-INs, and the conclusions largely rest on the idea that altered cell physiology is due to cell-autonomous loss of sodium channels in VIP-INs. The immunostaining has no negative control (is labeling lost/reduced in knockout mice?). Moreover, the labeling in Figure 1 looks completely non-specific. Accordingly, either the reagents should be validated, for example by examining staining in knockout mice, or the immunohistochemical results should be dropped from the study.

We agree with the reviewers’ conclusion that the Nav1.1 immunohistochemistry is sub-optimal and appreciate the reviewers’ suggestions for how to improve this data. We further agree that this data is critical to support the main finding of our paper. We have performed additional experiments to strengthen this data. Immunohistochemistry for Nav subunits and for proteins expressed at the axon initial segment is known to be extremely challenging, as discussed by others (e.g., Lorinz and Nusser, 2008, 2010; Schneider et al., 2006). We have adapted a sensitive, accurate, and reproducible method for labeling Nav subunits (Alshammari et al., 2016. Frontiers in Cellular Neuroscience). We have now included both positive and negative controls to support the specificity of our Nav1.1 immunohistochemistry. As suggested, we generated several litters of *Scn1a*-/- (null) mice via crossing *Scn1a*.VIP-Cre.tdT mice together and used the F1 null progeny as negative controls for the Nav1.1 immunohistochemistry. As has been published previously (Yu et al., 2006; Ogiwara et al., 2007; Mistry et al., 2014; etc.), these mice have a very severe phenotype and die by the age of ~P15. Therefore, we processed tissue from null mice at P13, at which time these mice already display prominent impairment including ataxia and spontaneous seizures. We used *n* = 3 *Scn1a*-/- as well as an additional cohort of *n* = 3 WT adult mice for Nav1.1 immunostaining as describe in the Materials and methods section. We examined two technical replicates of each mouse with both epifluorescence and confocal microscopy. There was no non-specific reactivity using the Neuromab Nav1.1 antibody in tissue from *Scn1a*-/- mice; however, there was strong Nav1.1 staining in tissue from WT mice processed in parallel (Figure 1—figure supplement 3).

In addition to this data, we also highlight the general staining pattern of Nav1.1 in neocortex using epifluorescence microscopy at lower magnification as well as confocal microscopy at high magnification. Nav1.1 staining is clearly most intense in layer 4 of the cortex, while it is somewhat sparser in layer 2/3 and 5, consistent with prominent expression in PV-INs and highest density of PV-INs in layer 4. In addition to the experiments requested by the reviewers, we undertook an additional positive control and performed staining in a PV-Cre.tdT mouse, showing high Nav1.1 expression on PV-IN axons. While staining on PV-INs appears to be higher than what we observed on VIP-IN axons, this is to be expected from the Allen brain institute single cell transcriptomics data which shows highest expression of Nav1.1 in PV-INs. This may be why Nav1.1 staining has been previously reported in PV-INs but was perhaps overlooked in VIP-INs. We are not arguing that Nav1.1 expression is higher in VIP-INs than in PV-INs; rather, we are reporting the new finding that VIP-INs express Nav1.1. We found it impossible to localize Nav1.1 signal specifically to VIP-IN axons using epifluorescence, but this became apparent using high power confocal imaging of fine structures. It is the case that Nav1.1 subunit immunohistochemistry on VIP-IN axons is not extremely robust, likely due to the inherent challenges of immunostaining for Nav subunits combined with the small (< 1 μm) caliber of VIP-IN axons.

2) The authors present experiments showing that the peptide toxin Hm1a, which destabilizes inactivation of Nav1.1 channels, enhances firing of IS-type VIP neurons while having less dramatic effects on firing of CA type VIP neurons and on PV cells, and with no effect on firing of pyramidal neurons. These experiments are very nice. However, it was very strange that the authors make no mention of the paper by Richards et al. (PNAS 2018,115(34):E8077-E8085) with very similar experiments with Hm1a on a different mouse model of Dravet syndrome, showing enhancement of firing of hippocampal GABAergic neurons but no effect on firing of hippocampal pyramidal neurons, and also in vivo experiments showing efficacy of the toxin to reduce seizures and mortality. The Richards paper also presents a more detailed analysis of the effects of Hm1a on the gating of Nav1.1 channels than in the Osteen papers, as well as more comprehensive data on selectivity for Nav1.1 over other Na channels and K channels. The original Osteen et al. characterization of Hm1a toxin shows some effect on Nav1.2 as well, leaving the conclusion of Nav1.1 expression less clear. In addition, the dosing used in this study appears to be much too high. In the Richards paper they showed that Nav1.1 is sensitive to 1 nM toxin, and they used 5 nM in their slice experiments. Here, Goff and Goldberg used 1 μM, which may well not be specific. Since this experiments is important for demonstrating that the Na current in VIP neurons is actually from Nav1.1, I then this result should be supported with an Hm1a concentration of ~5 nM.

We thank the reviewers for this observation. The Hm1a experiments reported in the original manuscript were performed prior to publication of the Richards et al., 2018 paper, and we based the concentration selected for use on data from the Osteen et al., 2016 paper, which first reported Hm1a as a Nav1.1 modulator. We did not intend to mislead the reader/reviewer by not citing the Richards et al. paper.

We agree that this “rescue” experiment with Hm1a is important, in that it shows the presence and functional expression of Nav1.1-containing Na^+^ channels in VIP-INs. Accordingly, we have repeated our findings in IS VIP-INs with 50 nM Hm1a (see below). We have also made sure to include a citation of Richards et al., 2018, as is appropriate.

In Osteen et al., 2016, the authors showed that 500 nm Hm1a showed high specificity for Nav1.1 in a heterologous system, with little effect on Nav1.2 and Nav1.3. As is commonly accepted in the field, higher concentrations of toxin are often required in the acute brain slice preparation to facilitate toxin penetration into the slice; hence, we chose 1 µM for these initial experiments. We further show that this concentration of Hm1a has no effect on the excitability of pyramidal neurons – which do not express Nav1.1 – supporting the specificity of Hm1a to Nav1.1 (although pyramidal neurons do express Nav1.2).

Richards et al., 2018, do report an EC50 value of 7.5 nM for the efficacy of Hm1a on area under the curve of the inward Na^+^ current, compared to a value of 38 nM reported by Osteen at al. (see Figure 1 in that paper). Richards et al., 2018, show no activity of Hm1a on Nav1.2, but instead show that Hm1a has some activity on Nav1.3 with an EC50 = 40 nM but with a max efficacy less than half of that observed for the action of Hm1a on Nav1.1. The authors find that “up to 50 nM Hm1a had no effect on other [Nav] subtypes.” This data was all produced using an automated planar patching system, and when the authors validated their finding with manual patching, they found that 5 nM Hm1a had no effect on the V1/2 of inactivation or on the tau of fast inactivation of Nav1.1, while 50 nM Hm1a impaired Nav1.1 inactivation as expected (Figure 2). They did not perform the same validation experiments for Nav1.3. Taken together, such data suggests that a concentration of 50 nM is specific for Nav1.1 in the acute brain slice preparation.

Finally, as discussed in the Richards et al., 2018 paper, the most sensitive non-Nav1.1 Nav1.X Na^+^ channel subunit appears to be Nav1.3, which is not present in mouse brain at older time points (> P18) (this is the “embryonic” Na^+^ channel subunit). Hence, the effects observed for Hm1a on VIP-INs in our experiments (which were done in P30+ mice) cannot be ascribed to a nonspecific action of Hm1a on Na^+^ channels containing Nav1.3 subunits. We were also reassured that Richards et al., 2018, found no activity of Hm1a on K^+^ currents. Hence, 50 nM Hm1a should approximate the effect of 1 µM on Nav1.1 inactivation, but with minimal to no activity on Nav1.3, and no activity on any other Nav1.X channels. We found that this concentration produced a full rescue of IS VIP-INs firing frequency that was nearly identical to results obtained with 1 µM Hm1a. The most likely non-Nav1.1 Nav1.X subunit found in VIP-INs based on data from the Allen brain institute is Nav1.2, and at this concentration there should be no non-specific activity of Hm1a on Na^+^ channels formed by this subunit. These results now more strongly support the hypothesis that Nav1.1 is present in VIP-INs, and that selective enhancement of Nav1.1 activity with Hm1a is sufficient to rescue the deficits in repetitive firing observed in VIP-INs in *Scn1a*^+/-^ mice. This in turn supports the notion that the observed deficits in VIP-INs in Scn1a^+/-^ mice are due to a cell-intrinsic loss of Nav1.1 leading to hypoexcitability of IS VIP-INs.

3) While the recordings are of good quality and many of the basic findings are reasonably well-supported, the overall presentation is confusing as to study goals. Is the focus about different classes of VIP-INs or about dysfunction in Dravet syndrome? This duality ultimately weakens the strength of conclusions in either case. The authors should address this issue in their revision.

We appreciate this comment and have attempted to clarify the focus and goals of this study. To be clear, the paper is about Dravet syndrome. The Title, Abstract, and Introduction, were written with the intent to reflect this. We have edited the manuscript to attempt to further highlight this focus.

In previous studies of Dravet Syndrome and *Scn1a*^+/-^ mice, less attention has been paid to the diversity of interneurons beyond the well-studied PV and SST subtypes. Dravet syndrome is a neurodevelopmental disorder defined by epilepsy, autism spectrum disorder, and intellectual impairment, and these entities are each thought to be due to neuronal circuit dysfunction involving impaired excitability of GABAergic interneurons. Hence, it is difficult to separate the topics of Dravet syndrome pathophysiology and interneuron function/diversity. We think that the identification of VIP-IN dysfunction may be a critical component to understanding Dravet syndrome pathogenesis based on the severe and durable cognitive deficits in Dravet syndrome patients and recently identified roles of VIP-INs in cognitive processing.

In this study, our investigations into mechanisms of circuit dysfunction in Dravet syndrome led us to the investigation of VIP-IN physiology, as our data appeared to segregate into two very clearly defined groups (IS and CA VIP-INs, with only IS VIP-INs being impaired in *Scn1a*^+/-^ mice). Hence, we were forced to reconcile the observation that some but not all VIP-INs were dysfunctional. This observation led us to study the basis of the IS firing pattern and the underlying biophysical basis of this difference, which revealed what we think is an interesting and important interaction between Na^+^ current density and M-current in these cells. We agree that there may be more discussion in the paper related to interneuron diversity than is necessary; that said, while we have attempted to further clarify the focus of the paper as noted above, it is not clear what of this interneuron diversity-related content, should be removed. We would respectfully suggest that our findings do add something to the field of interneuron diversity by providing a potentially useful classification of VIP-IN firing patterns and identifying two ion channels that regulate VIP-IN electrophysiological properties.

These two themes converge in the consideration of the role of neuromodulation in regulating cerebral cortical function and in circuit dysfunction in Dravet syndrome. M-current is a critical determinant of VIP-IN physiology, and cholinergic neuromodulation can specifically regulate IS VIP-INs through muscarinic acetylcholine receptor-mediated suppression of M-current.